# AdaSpec: Adaptive Spectrum for Enhanced Node Distinguishability

**Fangbing Liu & Qing Wang**
The Australian National University
`fangbing.liu,qing.wang@anu.edu.edu`

## Abstract

Spectral Graph Neural Networks (GNNs) achieve strong performance in node classification, yet their node distinguishability remains poorly understood. We analyze how graph matrices and node features jointly influence node distinguishability. Further, we derive a theoretical lower bound on the number of distinguishable nodes, which is governed by two key factors: distinct eigenvalues in the graph matrix and nonzero frequency components of node features in the eigenbasis. Based on these insights, we propose AdaSpec, an adaptive graph matrix generation module that enhances node distinguishability of spectral GNNs without increasing the order of computational complexity. We prove that AdaSpec preserves permutation equivariance, ensuring that reordering the graph nodes results in a corresponding reordering of the node embeddings. Experiments across eighteen benchmark datasets validate AdaSpec's effectiveness in improving node distinguishability of spectral GNNs. Code: https://github.com/Mia-321/AdaSpec

## 1 Introduction

Graph Neural Networks (GNNs) have become increasingly popular for graph learning tasks due to their strong performance in tasks such as graph and node classification (Kipf & Welling, 2017; Xu et al., 2019; He et al., 2021; Wang & Zhang, 2022; Qin et al., 2025). Among the various GNN models, spectral GNNs represent a prominent class that transforms graph signals into the spectral domain, enabling graph filters to process information for downstream tasks. Although numerous spectral GNNs have been proposed, their node distinguishability remains insufficiently understood. Node distinguishability refers to the capacity of a GNN to map topologically or feature-different nodes to different embeddings. These models typically utilize different graph matrices, such as the normalized adjacency or Laplacian matrix. Further, the distribution of node features across the graph plays a crucial role in model performance (He et al., 2022b; Platonov et al., 2023). To the best of our knowledge, no existing work has systematically analyzed the interaction between the graph matrix and node features in determining node distinguishability in spectral GNNs.

Spectral GNNs with state-of-the-art performance generally follow the form:

$$\Psi(M, X) = g_\Theta(M)f_W(X), \tag{1}$$

where $M \in \mathbb{R}^{n \times n}$ represents the graph matrix (such as the Laplacian or adjacency matrix), $X \in \mathbb{R}^{n \times h}$ denotes the node feature matrix, $g_\Theta(M) = \sum_{k=0}^{K} \theta_k T_k(M)$ is the graph convolution function parameterized by $\Theta = \{\theta_k\}_{k=0}^{K}$, and $T_k(\cdot)$ denotes the $k$-th polynomial basis. The term $f_W(X)$ represents the feature transformation function parameterized by $W$. Spectral GNNs learn meaningful node features by optimizing $W$, projecting them into the spectral domain. By adjusting $\Theta$, spectral GNNs filter out unnecessary information and enhance useful information for downstream tasks.

While this formulation illustrates how spectral GNNs process node features through graph convolution, their capacity for node distinguishability remains inadequately understood. This leads to a fundamental question: how does the interaction between the graph matrix $M$ and the node features $X$ projected into the spectral domain affect the node distinguishability of spectral GNNs? In this work, we demonstrate that node distinguishability is influenced by the eigenvalue multiplicity and the missing frequency components of node features in the eigenbasis of the graph matrix. Further, we derive a theoretical lower bound on the number of nodes that can be distinguished by spectral GNNs, given a specific graph matrix and node features.

Motivated by our theoretical analysis of node distinguishability, we introduce AdaSpec, an adaptive graph matrix generation module that optimizes the graph matrix to maximize its lower bound on node distinguishability. Designed as a plug-in, AdaSpec can be seamlessly integrated into any spectral GNN to enhance node distinguishability. Moreover, spectral GNNs augmented with AdaSpec preserve permutation equivariance, ensuring that reordering graph nodes results in a corresponding reordering of node embeddings. Finally, AdaSpec maintains the graph's connectivity, guaranteeing that the learned embeddings accurately reflect the underlying graph structure.

We evaluate our approach on eighteen benchmark node classification datasets, covering a range of small- and large-scale graphs with both homophilic and heterophilic structures in Section 6. Spectral GNNs with AdaSpec achieve notable performance improvements on heterophilic graphs, while maintaining or slightly improving accuracy on homophilic ones. These results validate the effectiveness of AdaSpec in boosting node distinguishability. Additionally, experimental results show that the order of time complexity of spectral GNNs with and without AdaSpec are the same.

## 2 Related Works

**Spectral GNNs.** Spectral GNNs perform graph convolution by applying filters in the spectral domain for representation learning. Based on the design of their graph filters, spectral GNNs can be categorized into polynomial (He et al., 2022a; 2021) and rational types (Levie et al., 2019; Bianchi et al., 2021; Li et al., 2025). Polynomial graph filters are computationally efficient and localized in the vertex domain (Hammond et al., 2009; Defferrard et al., 2016), and this paper focuses on their analysis. Recent studies primarily investigate how different polynomial bases affect spectral GNN performance, for instance, ChebNet, ChebNetII, JacobiConv, BernNet, GPRGNN and GLN (Defferrard et al., 2016; He et al., 2022a; Wang & Zhang, 2022; He et al., 2021; Chien et al., 2021; Li & Wang, 2024). Further, FavardGNN, UniFilter and PolyCF learn polynomial bases that adapt to different graph structures (Guo & Wei, 2023; Huang et al., 2024; Qin et al., 2025).

Above spectral GNNs use fixed graph matrices like normalized adjacency or Laplacian matrices. While research has focused on effect of polynomial bases on performance of spectral GNNs, we demonstrate the critical role of the graph matrix. We analyze how the interaction between the graph matrix and node features affects spectral GNN performance. Further, we propose AdaSpec, a graph matrix generation module to enhance the performance of spectral GNNs.

**Expressive Power of Spectral GNNs.** The expressive power of GNNs in graph classification has been extensively analyzed through the Weisfeiler-Lehman (WL) test (Li & Leskovec, 2022; Zhang et al., 2023; Jin et al., 2025), which are algorithms determining graph isomorphism (Weisfeiler & Leman, 1968). In contrast, the expressive power of GNNs for node classification remains less explored. The expressive capacity of linear spectral GNNs has been analyzed via the uniform approximation theorem in (Wang & Zhang, 2022), which shows that when the graph matrix has no repeated eigenvalues and node features span all frequency components, the model can approximate any one-dimensional function. However, these conditions rarely hold in real-world graphs, where symmetric structures are common and node features are often sparse. An eigenvalue correction method was proposed in (Lu et al., 2024) to enhance the expressiveness of spectral GNNs. This method reassigns eigenvalues purely by their sorted index, it does not preserve eigenspaces under node permutations, thereby breaking permutation equivariance, which is theoretically unsound.

Our work investigates the expressive power of spectral GNNs from the perspective of node distinguishability. We extend the understanding of how the interaction between the graph matrix and node features influences node distinguishability in spectral GNNs. Notably, our analysis goes beyond linear GNNs by incorporating nonlinear feature transformations $f_W$. Moreover, we rigorously establish a theoretical lower bound on the number of distinguishable nodes in spectral GNNs.

**Graph Rewiring.** Another line of works focuses on improving the performance of GNNs through graph rewiring techniques. Early methods include DropEdge and EDGEWIRE, which randomly remove edges to alleviate over-smoothing (Rong et al., 2020; Chan & Akoglu, 2016). Curvature-based approaches (Topping et al., 2022) adjust connectivity using discrete Ricci curvature to combat over-squashing, while locality-aware strategies preserve structures efficiency (Barbero et al., 2024). Recent methods include DiffWire, a differentiable and parameter-free approach guided by the Lovász bound (Arnaiz-Rodríéguez et al., 2022); FoSR, improving spectral expansion (Karhadkar et al., 2023); and GPER, selecting edges based on resistance to enhance information flow (Shen et al., 2024).

Objectives and underlying mechanisms of graph rewiring methods differ fundamentally from ours. Graph rewiring addresses structural issues by modifying graph topology in the spatial domain, our method enhances node distinguishability in the spectral domain. Our AdaSpec is not a competitor to graph rewiring. It is a plug-and-play spectral enhancement and can be seamlessly integrated with existing graph rewiring methods to achieve superior performance.

## 3 PRELIMINARIES

Let $G = (\mathcal{V}, \mathcal{E}, X)$ denote an undirected, simple graph, where $\mathcal{V}$ is the set of nodes with cardinality $|\mathcal{V}| = n$, $\mathcal{E}$ is the set of edges, and $X \in \mathbb{R}^{n \times h}$ is the node feature matrix. For each node $v \in \mathcal{V}$, $X(v) \in \mathbb{R}^h$ denotes its associated feature vector. The graph structure is represented by the adjacency matrix $A \in \{0,1\}^{n \times n}$, where $A_{ij} = 1$ if $(v_i, v_j) \in \mathcal{E}$, and 0 otherwise. The degree matrix $D \in \mathbb{R}^{n \times n}$ is diagonal with entries $D_{ii}$ equal to the degree of node $v_i$. The normalized adjacency matrix is defined as $\tilde{A} = D^{-\frac{1}{2}} A D^{-\frac{1}{2}}$. The normalized graph Laplacian is given by $\tilde{L} = I - \tilde{A}$, where $I \in \mathbb{R}^{n \times n}$ is the identity matrix.

Two nodes $u$ and $v$ in an undirected graph $G$ are *structurally equivalent* $s_u \sim s_v$ if they share exactly the same neighbors; formally, for every other node $w \in \mathcal{V} \setminus \{u, v\}$, $(u, w) \in \mathcal{E} \iff (v, w) \in \mathcal{E}$. In effect, swapping $u$ and $v$ leaves the graph's adjacency relation unchanged.

A *permutation* of the node set $\mathcal{V}$ is a bijection $\pi : \mathcal{V} \to \mathcal{V}$. The set of all permutations on $\mathcal{V}$ forms the symmetric group $\mathrm{Sym}(\mathcal{V})$. An *automorphism* of the graph $G$ is a permutation $\pi \in \mathrm{Sym}(\mathcal{V})$ satisfying the following conditions: (1) edge preservation: $(v, u) \in \mathcal{E} \iff (\pi(v), \pi(u)) \in \mathcal{E}$, $\forall v, u \in \mathcal{V}$, and (2) feature preservation: $X(\pi(v)) = X(v)$, $\forall v \in \mathcal{V}$. The *automorphism group* of $G$, denoted $\mathrm{Aut}(G)$, is the set of all such automorphisms.

Two nodes $u$ and $v$ are said to be *isomorphic*, denoted $u \sim v$, if they belong to the same orbit under $\mathrm{Aut}(G)$; that is, there exists a permutation $\pi \in \mathrm{Aut}(G)$ such that $\pi(v) = u$. Otherwise, $u$ and $v$ are *non-isomorphic*.

An important property of functions defined on graphs is *permutation equivariance*, which ensures that the output remains consistent under any reordering of the nodes. Formally,

**Definition 3.1** (Permutation Equivariance). Let $\mathcal{G}$ denote the set of graphs. A function $f : \mathcal{G} \to \mathbb{R}^{n \times d}$ is said to be *permutation equivariant* if, for any graph $G \in \mathcal{G}$ and any permutation $\pi \in \mathrm{Sym}(\mathcal{V})$, it holds that

$$f(\pi(G)) = \pi(f(G)),$$

where $\pi(G)$ denotes the graph obtained by permuting the nodes of $G$ according to $\pi$, and $\pi(f(G))$ denotes the corresponding permutation of the output of $f$.

## 4 NODE DISTINGUISHABILITY OF SPECTRAL GNNS

The node distinguishability of a spectral GNN refers to its ability to distinguish non-isomorphic nodes within graphs. Formally,

**Definition 4.1** (Node Distinguishability). For a spectral GNN with function class $\mathcal{F}$, where each $f \in \mathcal{F} : \mathcal{G} \to \mathbb{R}^{n \times d}$ maps a graph to node representations, node distinguishability refers to the ability to learn a function that assigns distinct representations to non-isomorphic nodes:

$$f(G)_v \neq f(G)_u \quad \text{for all } v, u \in \mathcal{V} \text{ where } v \not\sim u$$

where $f(G)_v$ and $f(G)_u$ denote representations of node $v$ and $u$. $v \not\sim u$ indicates node $u, v$ are non-isomorphic.

The spectral GNN's node distinguishability capacity that mapping non-isomorphic nodes to distinct representations is determined by its function class $\mathcal{F}$. To understand the distinguishability of spectral GNNs in the form of Equation (1) with input of graph matrix $M$ and feature matrix $X$, we begin by formally defining the spectrum of $M$ and the frequency components of $X$.

**Definition 4.2** (Spectrum and Frequency Components). Let $M = U \Lambda U^\top$ be the eigendecomposition of a graph matrix $M \in \mathbb{R}^{n \times n}$, where $\Lambda$ is a diagonal matrix of eigenvalues and $U = [u_1, \ldots, u_n]$ contains the corresponding eigenvectors. The *spectrum* of $M$, denoted $\mathrm{spec}(M)$, is the multiset of eigenvalues: $\mathrm{spec}(M) = \{\{\lambda_1, \lambda_2, \ldots, \lambda_n\}\}$, where $\lambda_i = \Lambda_{ii}$. Let support $\mathrm{supp}(\mathrm{spec}(M))$ be the underlying set of $\mathrm{spec}(M)$. Define $d_M = |\mathrm{supp}(\mathrm{spec}(M))|$, which is the number of distinct

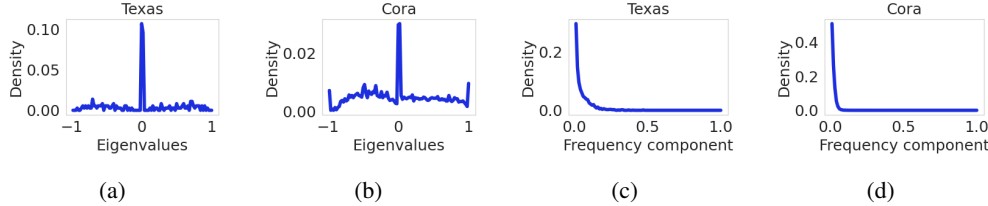

Figure 2: Eigenvalues and frequency component distributions.

eigenvalues. Given node features $X \in \mathbb{R}^{n \times h}$, the frequency components in the eigenbasis of $M$ are $\tilde{X} = U^\top X$, where $\tilde{X}_i = u_i^\top X$ is the $i$-th frequency component. The number of non-zero frequency components is $\|\tilde{X}^{(M)}\|_0 = |\{\tilde{X}_i \mid \tilde{X}_i \neq 0_h\}|$.

The limitations of node distinguishability in spectral GNNs stem from two key factors: Eigenvalue multiplicity of the graph matrix $M$ and the missing of frequency components of node features $X$ when projected onto the eigenbasis of $M$. In Figure 1, we show that spectral GNNs with a first-order polynomial filter and normalized adjacency matrix $\tilde{A}$ as graph matrix cannot distinguish node 1 and 6. (1) Non-distinguishable nodes can exist when there are missing frequency components that $d_{\tilde{A}} = 6 = n$ but $\|X^{(\tilde{A})}\|_0 = 5 < n$ in Figure 1(a). (2) Non-distinguishable nodes can exist when there are repeated eigenvalues $d_{\tilde{A}} = 3 < n$ even if $\|X^{(\tilde{A})}\|_0 = 6 = n$ in Figure 1(b). Nodes 1 and 6 in both subfigures are non-isomorphic but spectral GNNs yield identical embeddings for them. Hence they are indistinguishable. We provide a theoretical bound on the number of nodes that can be distinguished by spectral GNNs, stated as follows.

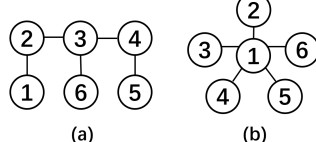

Figure 1: Nodes 1 and 6 cannot be distinguished by spectral GNNs of $K = 1$ with $\tilde{A}$ when graph signal $X = [1, 0, 0, 0, 0, 1]$. (a) Missing frequency components: $d_{\tilde{A}} = 6$, $\|X^{(\tilde{A})}\|_0 = 5$. (b) Eigenvalue multiplicity: , $d_{\tilde{A}} = 3$, $\|X^{(\tilde{A})}\|_0 = 5$.

**Theorem 4.3.** *For $X \neq 0_{n \times n}$, there exist a spectral GNN $\Psi(M, X)$ that can distinguish at least $\min(d_M, \|\tilde{X}^{(M)}\|_0)$ nodes on graph.*

This result provides a fundamental guarantee on the node distinguishability of spectral GNNs. The lower bound depends on both the number of distinct eigenvalues $d_M$ and the number of non-zero frequency components $\|\tilde{X}^{(M)}\|_0$, which together characterize the alignment between the graph matrix $M$ and the node features $X$. When multiple eigenvectors share the same eigenvalue, the graph filter $g_\Theta$ applies identical transformations to them, preventing from distinguishing different structural patterns. Similarly, if node features lack frequency components corresponding to certain eigenvectors, structural differences captured by those eigenvectors become invisible in embeddings. This has practical implications: increasing distinct eigenvlaue number $d_M$ and non-zero frequency components of $X$ in the eigenbasis of $M$ improves the theoretical guarantee on the lower bound of number of distinguishable nodes, offering a clear direction for enhancing the expressive power of spectral GNNs.

In real-world graphs, we observe that eigenvalue multiplicity and missing frequency component are very common.

**Observation I** (Eigenvalues of Multiplicity.) The normalized graph adjacency matrix $\tilde{A} = D^{-1/2} A D^{-1/2}$ often contains eigenvalues with multiplicities greater than one and the eigenvalue zero has largest multiplicity.

We illustrate the eigenvalue distribution of the normalized graph adjacency matrix for the Texas and Cora datasets in Figure 2(a-b). Additional eigenvalue distributions for various other real-world datasets are provided in Figure 3 (Appendix). This phenomenon is also observed in (Lim et al., 2023). Graph symmetry, repeated substructures often lead to repeated eigenvalues in the normalized adjacency matrix and reduce its rank. Real-world graphs also tend to be sparse due to many low-degree nodes, further lowering the rank. Since the rank of a real symmetric matrix equals the number of non-zero eigenvalues, low-rank matrices imply high multiplicity of the zero eigenvalue.

Node features in connected real-world graphs are sampled independently of the graph structure. For instance, in citation networks (such as Cora and PubMed), node features are the textual content of papers, which are collected independently of the graph structure. Thus, graph signals are not aligned with the graph's eigenvectors. We have below observations.

**Observation II** (Missing Frequency Components.) Many frequency components of graph signal (node feature) is zero in the eigenbasis of normalized graph adjacency matrix $\tilde{A}$.

We illustrate the distribution of frequency components for Texas and Cora in Figure 2(c-d), where most components are zero. Additional results for other real-world datasets are provided in Figure 4 (Appendix). Zero frequency component means that the frequency component in the direction of corresponding eigenvectors is missing. Real-world node features are often either smooth or oscillatory, containing only low or high-frequency components, leading to many others to be zero or negligible. Additionally, features are typically sparse, with only $k$ non-zero entries that $k \ll n$. When projected onto the eigenbasis, each component scales as $O(k/\sqrt{n})$. As $n \to \infty$, the proportion of non-zero frequency components tends toward zero.

Based on above observations and Theorem 4.3, we propose AdaSpec to enhance the node distinguishability of spectral GNNs.

## 5 ADASPEC

AdaSpec generates a graph matrix that adapts to both the graph structure and node features, enabling it to serve as a plug-in module for any spectral GNN $\Psi(M, X)$ of the form in Equation (1). The spectral GNN augmented with AdaSpec is defined as:

$$\Psi^+(A, X) = g_\Theta(\Omega(A, X))f_W(X), \tag{2}$$

where $\Omega$ maps the adjacency matrix $A$ and node features $X$ to a new graph matrix. The functions $g_\Theta$ and $f_W(X)$ remain the same as those in $\Psi(M, X)$.

AdaSpec enables $\Psi^+(A, X)$ to capture richer interactions between graph structure and node features, which are not possible using fixed matrices in classic spectral GNNs $\Psi(M, X)$. To ensure permutation equivariance of node embeddings, the generated graph matrix $M = \Omega(A, X)$ must satisfy two key properties: (1) $M$ commutes with $\mathrm{Aut}(G)$: $P_\sigma M = MP_\sigma, \forall \sigma \in \mathrm{Aut}(G)$ where $P_\sigma$ is the permutation matrix corresponding to the automorphism $\sigma$; (2) $M$ preserves edge connectivity: $M_{ij} \neq 0 \Leftrightarrow e_{ij} \in \mathcal{E}$ and $M_{ij} = 0 \Leftrightarrow e_{ij} \notin \mathcal{E}$. Thus, we design $\Omega(A, X)$ as

$$\Omega(A, X) = \Omega_D(A) + \alpha_1 \Omega_S(A) + \alpha_2 \Omega_F(X) \tag{3}$$

where $\Omega_D(A)$ is designed to increase the number of distinct eigenvalues, $\Omega_S(A)$ aims to reduce the multiplicity of zero eigenvalues, and $\Omega_F(X)$ is designed to decrease missing frequency components of $X$. The hyperparameters $\alpha_1, \alpha_2$ control the eigenvalue range for stable training.

### 5.1 INCREASE DISTINCT EIGENVALUES

According to Theorem 4.3, increasing the number of distinct eigenvalues of the graph matrix can raise the lower bound of number of nodes distinguished by a spectral GNN, thereby increasing its node distinguishability. To achieve this, the term $\Omega_D(A)$ in AdaSpec is designed as follows:

$$\Omega_D(A) = (D + B)^{-1/2} (A + B) (D + B)^{-1/2},$$

where $A$ and $D$ are the graph adjacency matrix and the degree matrix, respectively, and $B = \mathrm{diag}(b)$ is a learnable diagonal matrix with non-negative elements.

The diagonal element of $B$ is initialized as $b_u = 1/D_{uu}$, ensuring nodes with the same degree start with the same bias. For isomorphic nodes $u \sim v$, we have $b_u = b_v$ throughout training; for $u \nsim v$, training yields $b_u \neq b_v$. This initialization preserves permutation equivariance of $\Psi^+(A, X)$, as shown in Proposition 5.5. Adding $B$ to $A$ introduces node-specific flexibility, enabling $A + B$ and $D + B$ to adapt to graphs. This enhances node distinguishability by allowing structurally equivalent but feature different nodes to play distinct roles. For two non-isomorphic nodes $u, v$ that $u \nsim v$, if $s_u \sim s_v$ but $X(u) \neq X(v)$, introducing different biases $b_u \neq b_v$ breaks structure symmetry and reduces eigenvalue multiplicity. Intuitively, $B$ modifies the self-loop strength, altering information flow from the node itself. We later provide theoretical justification that this increases the number of distinct eigenvalues.

**Theorem 5.1** (Increased Distinct Eigenvalues). *Given a graph $G$ with the adjacency matrix $A$, and the degree matrix $D$, we have:*

$$d_{\Omega_D(A)} \geq d_{\tilde{A}}$$

We prove that for any $A$, there exist a diagonal matrix $B$ so that $\Omega_D(A)$ has $n$ distinct eigenvalues. This indicates that the lower bound of the number of distinguishable nodes for spectral GNNs using $\Omega_D$ is greater than or equal to that for those using $\tilde{A}$, according to Theorem 4.3.

## 5.2 Shifts Eigenvalues From Zero

The presence of zero eigenvalues forces spectral filters to suppress the associated frequency components, thereby hindering node distinguishability. We shift eigenvalues away from zero by using:

$$\Omega_S(A) = I.$$

We choose the identity matrix because adding it to any matrix shifts the eigenvalues while preserving the eigenvectors. This ensures minimal alteration to the original matrix.

Adding term $\epsilon \Omega_S$ to any matrix $C$ can reduce the number of zero eigenvalues. As all eigenvalues of $C$ add the same scalar $\epsilon$, distinct eigenvalues remain distinct after addition. As all eigenvectors of $C$ stays the same, so the number of non-zero frequency component of node feature stays the same.

## 5.3 Increase Frequency Components

We can increase the number of non-zero frequency component to the node distinguishability of spectral GNNs. Given a node feature matrix $X$, we design a matrix $\Omega_F$ that adapts to $X$ to increase the frequency components:

$$\Omega_F(X) = \sum_{i=1}^{h} \frac{X_{:i} X_{:i}^{\top}}{\|X_{:i}\|_F^2} \circ A \tag{4}$$

where $\circ$ denotes the Hadamard product.

By dividing by the Frobenius norm $\|X_{:i}\|_F^2$, features with larger magnitudes don't dominate the transformation. We prove in theory that for any symmetric matrix $C$ of no repeated eigenvalues, adding $\epsilon \Omega_F(X)$ can increase non-zero frequency components.

**Theorem 5.2** (Non-Decreasing Frequency Components). *For a real symmetric matrix $C \in \mathbb{R}^{n \times n}$ of no repeated eigenvalues with orthonormal basis $\{u_r\}_{r \in [n]}$. Under Condition 5.3, the following holds for index $i \in [h]$:*

$$\|\tilde{X}_{:i}^{(C+\epsilon \Omega_F)}\|_0 > \|\tilde{X}_{:i}^{(C)}\|_0$$

*where $\epsilon$ is a non-zero constant.*

**Condition 5.3** (Non-zero feature projections). Let $C \in \mathbb{R}^{n \times n}$ be a real symmetric matrix with orthonormal eigenbasis $\{u_r\}_{r=1}^{n}$. There exist two column node feature vectors $X_{:i}$ and $X_{:l}$ with $i, l \in [h]$ and $i \neq l$ such that $u_k^{\top} X_{:i} \neq 0$, $u_k^{\top} X_{:l} \neq 0$, and $u_j^{\top} X_{:l} \neq 0$ for some indices $k, j \in [n]$.

Condition 5.3 are naturally satisfied in most real-world graph datasets. This condition requires that node features have non-zero projections onto certain eigenvectors of the graph matrix. Natural heterogeneity in node features makes it likely that different nodes will have diverse nonzero projections onto eigenvectors, even with sparse features. Additionally, while feature correlation exists, real-world graph typically varies a lot along certain dimensions, satisfying our non-zero projection condition. Therefore, incorporating $\Omega_F(X)$ ensures that the number of non-zero frequency components of node features is increased in real-world graphs.

In summary, each component of $\Omega(A, X)$ either increases the number of distinct eigenvalues or the number of non-zero frequency components of the node features in the eigenbasis of the graph matrix. By Theorem 4.3, this leads to a higher lower bound on the number of distinguishable nodes, thereby enhancing node distinguishability. We show properties of our design $\Omega(A, X)$ as below.

**Theorem 5.4.** *For a graph $G$, the learnable matrix $\Omega(A, X)$ is commutative with $\mathrm{Aut}(G)$ and preserves edge connectivity.*

As $\Omega(A, X)$ satisfies desirable properties, it ensures that the augmented spectral GNNs $\Psi^+(A, X)$ with AdaSpec remains permutation equivariant.

**Proposition 5.5.** *When $f_W$ is permutation equivariant, spectral GNNs $\Psi^+(A, X)$ augmented with AdaSpec is permutation equivariant.*

Theorem 5.4 and Proposition 5.5 ensures that for spectral GNNs $\Psi^+(A, X)$, reordering the graph nodes results in a corresponding reordering of node embeddings. AdaSpec can be combined with any spectral GNNs to enhance their node distinguishability.

### 5.4 Time Complexity Analysis

The time complexity of classic spectral GNNs $\Psi(M, X)$ and $\Psi^+(A, X)$ augmented with AdaSpec is in the same order in both forward and backward propagation. $\Omega_F(X)$ in AdaSpec will increase the pre-computing time, but it needs to be computed only once. We list the time complexity in Table 1.

The time complexity can be analyzed in two main phases: pre-computation and forward/backward propagation. During pre-computation, graph matrix normalization requires $O(|\mathcal{V}| + |\mathcal{E}|)$ operations such as graph adjacency matrix normalization. $\Omega_F(X)$ in $\Psi^+(A, X)$ requires an additional $O(h(|\mathcal{V}| + |\mathcal{E}|))$ where computation is efficiently limited to non-zero entries in the adjacency matrix. Thus, the one-off pre-computing of $\Psi^+(A, X)$ scales linearly in the size of graph and node feature dimension.

For forward and backward propagation, the feature transformation step $f_W(X)$ incurs a complexity of $O(|W|h)$, while graph convolution $g_\Theta$ requires $O(KT|\mathcal{E}|)$ operations when $T_k(M)$ is computed recursively, such as in ChebNet, JacobiConv. Although $\Psi^+(A, X)$ requires additional computation of $\Omega(A, X)$ during each forward pass and gradient calculation for matrix $B$ during backpropagation at a cost of $O(|\mathcal{V}| + |\mathcal{E}|)$, this does not change the overall asymptotic complexity.

## 6 Experiments

We design our experiments to investigate the following research questions: (1) **Q1:** To what extent does AdaSpec generate task-adaptive graph matrices that enhance node distinguishability in spectral GNNs? (2) **Q2:** What is the contribution of each component within AdaSpec to overall performance? (3) **Q3:** How does AdaSpec affect the spectral properties of the graph matrix, particularly in terms of increasing the number of distinct eigenvalues? (4) **Q4:** What is the computational overhead introduced by integrating AdaSpec into spectral GNNs during training?

**Experimental Setup.** We conduct experiments on eighteen benchmark datasets for node classification to verify the effectiveness of AdaSpec. Datasets includes: six small heterophilic graphs (Texas, Wisconsin, Actor, Chameleon, Squirrel, Cornell), five large heterophilic graphs (Roman_Empire, Amazon_Ratings, Minesweeper, Tolokers, Questions) and seven homophilic graphs (Citeseer, Pubmed, Cora, Computers, Photo, Coauthor-CS, Coauthor-Physics). Statistics of datasets, details about the baselines, and the setting of hyperparameters are included in Appendix B. For each dataset, we follow (Chien et al., 2021; He et al., 2022a) and use sparse splitting that nodes are randomly divided into training/validation/testing with ratios of $2.5\%/2.5\%/95\%$, respectively. Notably, for Citeseer, Pubmed, and Cora datasets, 20 nodes per class are for training, 500 nodes for validation, and 1,000 nodes for testing.

We chose five popular spectral GNNs as our baselines: ChebNet (Defferrard et al., 2016), GPRGNN (Chien et al., 2021), BernNet (He et al., 2021), JacobiConv (Wang & Zhang, 2022), and ChebNetII (He et al., 2022a), and compare their performances augmented with AdaSpec and with fixed graph matrix across all datasets. For each spectral GNN, we use GNN (O) to denote the original model and GNN (M) to denote the spectral GNNs augmented by AdaSpec, with $\Delta \uparrow$ indicating the performance improvement.

**Effectiveness of AdaSpec.** We present the node classification performance with and without the AdaSpec on all small heterophilic datasets and a subset of large heterophilic datasets in Table 2. The Minesweeper and Question datasets are particularly challenging to classify, as their label informativeness (i.e., the mutual information between the labels of the central node and its neighbors) is zero (Platonov et al., 2023). The complete experimental results are in Table 9 (Appendix). Results on homophilic graphs are shown in Table 3 .

| Spectral GNNs | Parameter Count | Pre-computing Complexity | Forward/Backward Complexity |
|---|---|---|---|
| $\Psi(M, X)$ | $1 + K$ | $O(|\mathcal{V}| + |\mathcal{E}|)$ | $O(KT|\mathcal{E}| + |\mathcal{V}||W|)$ |
| $\Psi^+(A, X)$ | $1 + K + |\mathcal{V}|$ | $O(h(|\mathcal{V}| + |\mathcal{E}|))$ | $O(KT|\mathcal{E}| + |\mathcal{V}||W|)$ |

Table 1: Time complexity comparison of GNNs with and without AdaSpec. $\mathcal{V}$ and $\mathcal{E}$ denotes the node and edge set respectively. $h$ is the node feature dimension. $T$ is the node class number. $K$ is the polynomial order of spectral GNNs.

| Model | Texas | Wisconsin | Actor | Chameleon | Squirrel | Cornell | Minesweeper | Questions |
|---|---|---|---|---|---|---|---|---|
| ChebNet(O) | 38.67±9.31 | 32.92±7.38 | 25.15±0.69 | 29.32±4.13 | 24.23±3.24 | 31.33±7.51 | 86.29±0.2 | 55.13±0.54 |
| ChebNet(M) | 51.16±8.56 | 33.83±9.38 | 25.38±0.67 | 29.73±3.3 | 23.2±3.94 | 33.47±7.92 | 86.7±0.23 | 55.2±1.52 |
| Δ ↑ | +12.49 | +0.91 | +0.23 | +0.41 | -1.03 | +2.14 | +0.41 | +0.07 |
| ChebNetII(O) | 56.24±1.39 | 51.5±5.63 | 29.89±0.68 | 35.26±3.66 | 37.19±0.66 | 39.54±6.88 | 78.35±0.14 | 64.13±0.95 |
| ChebNetII(M) | 56.76±3.12 | 52.0±7.75 | 30.43±1.23 | 35.62±3.52 | 36.88±0.69 | 39.94±7.05 | 79.1±0.09 | 65.54±0.7 |
| Δ ↑ | +0.52 | +0.5 | +0.54 | +0.36 | -0.31 | +0.4 | +0.75 | +1.41 |
| JacobiConv(O) | 55.09±5.95 | 49.0±10.51 | 32.15±0.77 | 34.29±3.82 | 29.29±1.99 | 38.96±8.79 | 87.34±0.12 | 64.72±0.38 |
| JacobiConv(M) | 57.4±3.93 | 52.33±8.88 | 32.52±0.75 | 38.16±1.18 | 31.35±1.68 | 41.62±10.06 | 89.13±0.1 | 65.8±0.18 |
| Δ ↑ | +2.31 | +3.33 | +0.37 | +3.87 | +2.06 | +2.66 | +1.79 | +1.08 |
| GPRGNN(O) | 48.15±4.74 | 44.25±5.92 | 30.39±1.24 | 32.5±2.92 | 27.7±3.88 | 34.39±6.88 | 87.15±0.49 | 53.14±0.27 |
| GPRGNN(M) | 58.27±4.97 | 53.25±7.21 | 30.4±1.51 | 32.82±4.76 | 27.3±6.03±4.77 | 36.13±7.52 | 88.58±0.18 | 58.19±0.36 |
| Δ ↑ | +10.12 | +9.0 | +0.01 | +0.32 | -0.4 | +1.74 | +1.43 | +5.05 |
| BernNet(O) | 56.19±7.52 | 49.38±5.75 | 30.5±1.18 | 35.35±3.46 | 33.41±3.42 | 36.82±10.64 | 76.54±0.23 | 64.86±0.37 |
| BernNet(M) | 58.9±4.11 | 51.96±7.84 | 30.61±0.67 | 39.61±1.55 | 34.46±3.52 | 40.23±5.66 | 76.95±0.21 | 65.2±0.31 |
| Δ ↑ | +2.71 | +2.58 | +0.11 | +4.26 | +1.05 | +3.41 | +0.41 | +0.34 |

Table 2: Performance of spectral GNNs with/without AdaSpec on heterophilic datasets. ROC AUC is reported on Minesweeper, Questions. Testing accuracy is reported on other datasets. High accuracy and ROC AUC indicate good performance.

| Model | Citeseer | Pubmed | Cora | Computers | Photo | Coauthor-CS | Coauthor-Physics |
|---|---|---|---|---|---|---|---|
| ChebNet(O) | 69.21±0.87 | 75.29±2.34 | 80.45±1.09 | 82.64±1.76 | 91.77±0.32 | 90.95±0.34 | 95.03±0.11 |
| ChebNet(M) | 68.52±0.86 | 77.38±1.45 | 82.26±0.84 | 85.14±0.89 | 92.34±0.41 | 91.54±0.22 | 94.93±0.09 |
| Δ ↑ | -0.69 | +2.09 | +1.81 | +2.5 | +0.57 | +0.59 | -0.1 |
| ChebNetII(O) | 69.93±1.15 | 78.42±1.48 | 81.64±0.86 | 84.96±0.97 | 92.71±0.46 | 93.08±0.27 | 95.23±0.1 |
| ChebNetII(M) | 69.54±0.9 | 78.59±1.52 | 81.97±0.86 | 84.79±0.83 | 92.58±0.31 | 93.11±0.25 | 95.26±0.11 |
| Δ ↑ | -0.39 | +0.17 | +0.33 | -0.17 | -0.13 | +0.03 | +0.03 |
| JacobiConv(O) | 70.8±0.7 | 79.43±1.45 | 77.15±0.96 | 85.39±0.95 | 92.79±0.38 | 93.33±0.23 | 95.32±0.15 |
| JacobiConv(M) | 70.91±0.66 | 79.65±1.25 | 83.52±0.69 | 84.92±0.92 | 92.83±0.36 | 93.27±0.25 | 95.43±0.11 |
| Δ ↑ | +0.11 | +0.22 | +6.37 | -0.47 | +0.04 | -0.06 | +0.11 |
| GPRGNN(O) | 70.02±0.7 | 79.24±1.1 | 82.24±0.86 | 84.09±0.81 | 92.43±0.24 | 92.99±0.22 | 95.28±0.04 |
| GPRGNN(M) | 70.4±0.41 | 79.6±0.97 | 82.19±0.79 | 84.28±0.86 | 92.53±0.38 | 93.33±0.29 | 95.32±0.15 |
| Δ ↑ | +0.38 | +0.36 | -0.05 | +0.19 | +0.1 | +0.34 | +0.04 |
| BernNet(O) | 69.12±0.96 | 78.9±1.04 | 81.9±0.8 | 85.15±1.14 | 92.63±0.29 | 93.11±0.23 | 95.3±0.17 |
| BernNet(M) | 69.45±0.64 | 79.07±1.03 | 82.5±0.78 | 85.18±0.77 | 92.58±0.36 | 93.07±0.29 | 95.32±0.15 |
| Δ ↑ | +0.33 | +0.17 | +0.6 | +0.03 | -0.05 | -0.04 | +0.02 |

Table 3: Test accuracy of spectral GNNs with/without AdaSpec on homophilic datasets. High accuracy indicates good performance.

From Tables 2 and 3, we observe the following: (1) AdaSpec significantly improves performance on heterophilic graphs compared to homophilic graphs. There is an average accuracy improvement of 1.89% on small heterophilic graphs, an average ROC AUC improvement of 1.27% on large heterophilic graphs, and an average accuracy improvement of 0.43% on homophilic graphs. (2) AdaSpec shows greater performance improvement on small-sized graphs compared to large-sized graphs. The average node classification accuracy improvement on small graphs (Texas, Wisconsin, Cornell) is 3.45%, whereas the improvement on larger graphs (Chameleon, Squirrel) is 0.46%.

The main performance improvement stems from AdaSpec's ability to increase node distinguishability in spectral GNNs. By refining the graph structure representation, AdaSpec enables the model to better separate nodes with similar features or structures. In homophilic graphs, low-frequency components are sufficient for smooth features, so adding more may hurt. Heterophilic graphs require richer spectral patterns, and AdaSpec help by increasing useful frequency components. In small graphs, changes in graph matrix can reveal critical structure. In large graphs, existing structure dominates, changes in graph matrix are less effective.

**Component-wise Analysis.** We report ChebNet performance augmented with AdaSpec across multiple datasets and conduct an ablation study to isolate the effects of each component. Results in Table 4 show: (1) Full components: Combining all three components consistently yields the best performance. (2) Structure-dominated graphs (e.g., Chameleon, Cora): $\Omega_D$ outperforms $\Omega_S$. (3) Feature-dominated graphs (e.g., Texas, Roman_Empire): $\Omega_S$ outperforms $\Omega_D$. (4) Frequency components: Increasing non-zero frequency components via $\Omega_F(X)$ improves performance, even when used alone. Each component within AdaSpec independently improves node distinguishability. When combined, these mechanisms complement each other, leading to the strongest overall performance.

| AdaSpec | Texas | Chameleon | Roman Empire | Amazon Ratings | Citeseer | Cora |
|---|---|---|---|---|---|---|
| ChebNet(O) | 38.67 | 29.32 | 47.15 | 39.79 | 69.21 | 80.45 |
| $\Omega_D(A)$ | 40.75 | 26.71 | 22.70 | 40.75 | 68.27 | 81.53 |
| $\Omega_S(A)$ | 44.51 | 23.27 | 54.04 | 35.28 | 52.29 | 55.63 |
| $\Omega_F(X)$ | 26.24 | 28.22 | 54.12 | 37.16 | 29.49 | 65.49 |
| $\Omega(A,X)$ | 51.16 | 29.73 | 54.55 | 40.92 | 68.52 | 82.26 |

Table 4: Test accuracy of ChebNet with different components of AdaSpec across datasets that $\Omega(A,X)$ contains all three components.

**Increased Distinct Eigenvalue Number.** We compare the number of distinct eigenvalues between the original normalized adjacency matrix $\tilde{A}$ and the modified matrix $\Omega_D(A)$ from AdaSpec when using ChebNet. Due to the computational cost of full eigendecomposition, we conduct this analysis on small-scale homophilic and heterophilic datasets. As shown in Table 5, $\Omega_D(A)$ consistently increases the number of distinct eigenvalues, supporting Theorem 5.1. Standard normalized adjacency matrix $\tilde{A}$ and its self-loop version $\hat{A}$ are specific cases of the component $\Omega_D(A)$ in AdaSpec by setting $B = 0$ and $B = 1$ respectively. We introduces richer structural information in spectral GNNs by making $B$ learnable matrix (updated via gradient descent) in AdaSpec. The increased number of distinct eigenvalues directly enhances the model's ability to differentiate non-isomorphic nodes.

| Dataset | Texas | Wisconsin | Chameleon | Squirrel | Cornell | Citeseer | Cora |
|---|---|---|---|---|---|---|---|
| $|\mathcal{V}|$ | 183 | 251 | 890 | 2,223 | 183 | 3,327 | 2,708 |
| $d_{\tilde{A}}$ | 113 | 178 | 845 | 2,213 | 122 | 2,508 | 2,395 |
| $d_{\Omega_D(A)}$ | 181 | 229 | 888 | 2,221 | 144 | 3,227 | 2,645 |
| $\triangle \uparrow$ | 68 | 51 | 43 | 8 | 22 | 719 | 250 |

Table 5: Number of distinct eigenvalues of the graph matrix. $|\mathcal{V}|$ denotes the number of nodes in graphs. $d_{\tilde{A}}$ and $d_{\Omega_D(A)}$ are numbers of distinct eigenvalues of $\tilde{A}$ and $\Omega_D(A)$ in AdaSpec respectively.

**Time Complexity of AdaSpec.** We evaluate the training efficiency of ChebNet with and without AdaSpec across multiple datasets. For each dataset, we conduct ten independent runs. We report the average training time per run and the pre-computing time of $\Psi^+(A,X)$ in Table 6. The results show that AdaSpec introduces minimal overhead and can even accelerate convergence on large heterophilic graphs (e.g., Roman_Empire, Amazon_Ratings). When increase graph size from Amazon_Ratings to Coauthor-Physics, the pre-computation time rises from 0.03s to 12.44s, which is consistent with our time complexity analysis in Section 5.4. By incorporating structural and feature bias into the node representation, AdaSpec enables faster convergence and more efficient training.

| Datasets | Roman _Empire | Amazon _Ratings | Tolokers | Minesweeper | Questions | Computers | Photo | Coauthor -CS | Coauthor -Physics |
|---|---|---|---|---|---|---|---|---|---|
| ChebNet (O) | 1.93 | 1.91 | 1.76 | 1.28 | 2.53 | 4.73 | 3.4 | 3.67 | 4.54 |
| ChebNet (M) | 1.88 | 1.35 | 2.51 | 2.18 | 3.05 | 5.32 | 4.83 | 4.11 | 4.60 |
| $\Delta \uparrow$ | -0.05 | -0.56 | 0.75 | 0.9 | 0.52 | 0.59 | 1.43 | 0.44 | 0.06 |
| Pre-Computing | 0.26 | 0.03 | 0.44 | 0.08 | 0.56 | 1.83 | 0.9 | 4.1 | 12.44 |

Table 6: Average training and pre-computing time (in seconds) for ChebNet with and without AdaSpec on large heterophilic and homophilic datasets. Pre-computing is for $\Omega_F(X)$ in AdaSpec.

## 7 CONCLUSION AND LIMITATIONS

This work analyzes node distinguishability of spectral GNNs and shows it is governed by the interplay between the graph matrix and node features. Specifically, by the number of distinct eigenvalues and nonzero frequency components in the graph matrix's eigenbasis. We propose AdaSpec, a plug-in module that enhances the node distinguishability of spectral GNNs, offering theoretical guarantees and empirical gains.

While effective, our approach is limited to spectral GNNs and provides only a lower bound on distinguishability. The design of AdaSpec is tailored to certain data distributions and may not generalize universally. Future work could explore more generalizable graph matrix designs, applications to dynamic graphs, and integration with advanced spectral GNNs for broader applicability.

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

APPENDIX

## A PROOFS

Detailed proofs of theorems and propositions are provided.

**Theorem 4.3.** *For $X \neq 0_{n \times n}$, there exist a spectral GNN $\Psi(M, X)$ that can distinguish at least $\min(d_M, \|\tilde{X}^{(M)}\|_0)$ nodes on graph.*

*Proof.* (1) Rank of $f_W$.

The rank of a matrix corresponds to the dimension of its column space. When $f_W$ is MLP, which can approximate any function, there exist a parameter $W'$ so that $f_{W'}$ is injective function, and $\text{rank}(f_{W'}(X)) = \text{rank}(X)$.

(2) Rank of $g_\Theta(M)$.

For $K$ order polynomial function on symmetric graph matrix $g(M)$, we can represent it as $g(M) = \sum_{k=0}^K \alpha_k M^k$. We conduct eigendecomposition $M = U\Lambda U^T$, thus, $g(M) = Ug(\Lambda)U^T$, where $g(\lambda_i) = \sum_{k=0}^K \alpha_k \lambda_i^k$. $Rank(g(M))$ equals the number of non-zeros in $g(\Lambda)$. When $\alpha_0 \neq 0$, we have $Rank(g(M)) = n$ as $I$ is full rank matrix. Therefore, there exist a parameter $\Theta'$ that $\theta_0' \neq 0$, such that $\text{rank}(g_{\Theta'}(M)) = n \geq \text{rank}(M)$.

(3) Rank and eigenvalues.

As $g_{\Theta'}(M)$ is a full rank matrix, so $\text{rank}(g_{\Theta'}(M)) \geq \text{rank}(M) \geq d_M$.

As eigenvectors of $M$ are linearly independent, if $X$ has $r$ non-zero frequency components, then there at least $r$ linearly independent directions to represent $X$ in eigenbasis of $M$, i.e., $\text{rank}(X) \geq \|\tilde{X}^{(M)}\|_0$.

Thus, for spectral GNN $\Psi$ in Equation (1), there exist a parameter $\Theta', W'$ that

$$
\begin{aligned}
&\text{rank}(\Psi(M, X)) \\
&= \text{rank}(g_{\Theta'}(M)f_{W'}(X)) \\
&\geq \min(\text{rank}(g_{\Theta'}(M)), \text{rank}(f_{W'}(X))) \\
&\geq \min(d_M, \text{rank}(X)) \\
&\geq \min(d_M, \|\tilde{X}^{(M)}\|_0)
\end{aligned}
\tag{5}
$$

If $\text{rank}(\Psi(M, X)) \geq r$, it means that at least $r$ rows in embeddings $\Psi(M, X)$ are linearly independent. Thus, $\Psi(M, X)$ can distinguish $r$ nodes in graph.

In summary, there exist a spectral GNN that can distinguish at least $\min(d_M, \|\tilde{X}^{(M)}\|_0)$ on graph. $\square$

**Lemma A.1** (First-Order Eigenvalue Perturbation (Kato, 2013)). *Let $\Omega(t)$ be an analytic family of real symmetric matrices, and let $\lambda$ be an eigenvalue of $\Omega(0)$ with multiplicity $m$. Let $V_\lambda \in \mathbb{R}^{n \times m}$ have orthonormal columns spanning the eigenspace of $\lambda$. Then there exist real-analytic eigenvalue branches $\lambda_1(t), \ldots, \lambda_m(t)$ of $\Omega(t)$ with $\lambda_i(0) = \lambda$, and their first-order derivatives at $t = 0$ satisfy*

$$
\lambda_i'(0) = \mu_i,
$$

*where $\mu_1, \ldots, \mu_m$ are the eigenvalues of the compressed matrix*

$$
H_\lambda = V_\lambda^\top \left. \frac{d}{dt}\Omega(t) \right|_{t=0} V_\lambda.
$$

*In particular, the multiplicity-$m$ eigenvalue $\lambda$ splits to first order precisely when $H_\lambda$ is not a scalar multiple of the identity.*

**Theorem 5.1** (Increased Distinct Eigenvalues). *Given a graph $G$ with the adjacency matrix $A$, and the degree matrix $D$, we have:*

$$
d_{\Omega_D(A)} \geq d_{\tilde{A}}
$$

*Proof.* When $\tilde{A}$ has an eigenvalue $\lambda$ of multiplicity $k > 1$ and let $V_\lambda \in \mathbb{R}^{n \times k}$ have orthonormal columns spanning the corresponding eigenspace $E_\lambda$.

Consider the analytic family

$$\Omega(t) := (D + tB)^{-1/2}(A + tB)(D + tB)^{-1/2}, \qquad t \in [0, t_0],$$

so $\Omega(0) = \tilde{A}$. The first-order perturbation matrix at $t = 0$ is

$$P := \frac{d}{dt}\Omega(t)\Big|_{t=0} = -\tfrac{1}{2}D^{-3/2}BAD^{-1/2} + D^{-1/2}BD^{-1/2} - \tfrac{1}{2}D^{-1/2}AD^{-3/2}B.$$

Equivalently, conjugating by $D^{1/2}$ yields the simpler form

$$Q := D^{1/2}PD^{1/2} = B - \tfrac{1}{2}\big(D^{-1}B\,A + A\,D^{-1}B\big),$$

from which $P = D^{-1/2}QD^{-1/2}$.

$B = diag(b) = (b_1, \ldots, b_n)$ and $Q$ depends linearly on the diagonal vector $b$, hence so does $P$.

By Lemma A.1, the first-order shifts of the $k$ eigenvalue branches emanating from $\lambda$ are the eigenvalues of the $k \times k$ symmetric matrix

$$H_\lambda \;=\; V_\lambda^\top P V_\lambda \;=\; V_\lambda^\top D^{-1/2}\,Q\,D^{-1/2}V_\lambda.$$

Since $Q$ and $H_\lambda$ is linear in $b$, the condition that $H_\lambda$ be a scalar matrix is a finite system of homogeneous linear equations in $b$. Thus the "bad set"

$$\mathcal{S}_\lambda := \{b \in \mathbb{R}^n : H_\lambda \text{ is scalar}\}$$

is a proper linear subspace of $\mathbb{R}^n$ (for $k > 1$ the system is nontrivial unless the eigenspace has a special coordinate structure). Taking the finite union over all repeated eigenvalues produces a proper algebraic subset $\mathcal{S} = \bigcup_\lambda \mathcal{S}_\lambda \subset \mathbb{R}^n$.

Choose $b^* \notin \mathcal{S}$ (and $b_i^* > 0$). Then for each repeated eigenvalue $\lambda$ the corresponding $H_\lambda$ is non-scalar, so the multiplicity of $\lambda$ splits into at least two distinct eigenvalue branches for small $t > 0$. Therefore for sufficiently small $t > 0$ the matrix $\Omega(t)$ has strictly more distinct eigenvalues than $\tilde{A}$. Setting $B^* = t \operatorname{diag}(b^*)$ yields the desired diagonal perturbation.

Because we have produced a point outside the discriminant variety, the discriminant polynomial is not identically zero; hence the complement (the set of $B$ giving simple spectrum) is Zariski-open and dense in $\mathbb{R}^n$.

In summary, there exists a diagonal $B^*$ (indeed a Zariski-open dense set of such diagonals) for which $\Omega_D(A) = (D + B^*)^{-1/2}(A + B^*)(D + B^*)^{-1/2}$ has all eigenvalues simple, i.e., $d_{\Omega_D(A)} = n \geq d_{\tilde{A}}$.

$\square$

**Theorem A.2** (First-order Perturbation Theorem (Stewart, 1990))**.** *When a system described by a matrix $A \in \mathbb{R}^{n \times n}$ of no repeated eigenvalues is slightly altered by a small perturbation $\zeta \in \mathbb{R}^{n \times n}$ and the new new system can be represented as $A' = A + \epsilon\zeta$, where $\epsilon$ is a non-zero constant. $A$ has eigenvalues $\{\lambda_i\}_{i \in [n]}$ and eigenvectors $\{u_i\}_{i \in [n]}$. $A'$ has eigenvalues $\{\lambda_i'\}_{i \in [n]}$ and eigenvectors $\{u_i'\}_{i \in [n]}$.*

*Relations between eigenvalues and eigenvectors of $A$, $A'$ are:*

$$\lambda_i' = \lambda_i + \epsilon\delta\lambda_i = u_i^\top \zeta u_i + O(\epsilon^2)$$

$$u_i' = u_i + \epsilon \sum_{j \neq i} \frac{u_j^\top \zeta u_i}{\lambda_i - \lambda_j} u_j + O(\epsilon^2)$$

**Theorem 5.2** (Non-Decreasing Frequency Components)**.** *For a real symmetric matrix $C \in \mathbb{R}^{n \times n}$ of no repeated eigenvalues with orthonormal basis $\{u_r\}_{r \in [n]}$. Under Condition 5.3, the following holds for index $i \in [h]$:*

$$\|\tilde{X}_{:i}^{(C+\epsilon\Omega_F)}\|_0 > \|\tilde{X}_{:i}^{(C)}\|_0$$

*where $\epsilon$ is a non-zero constant.*

*Proof.* Since $C$ is a real symmetric matrix, it can be diagonalized

$$C = U\Lambda U^T$$

where $U = [u_1, \ldots, u_n]$ is orthonormal eigenvectors and $\Lambda = \text{diag}(\lambda_1, \ldots, \lambda_n)$ is the diagonal matrix of eigenvalues.

We denote $\{\tilde{\lambda}_i\}_{i \in [n]}$ and $\{\tilde{u}_i\}_{i \in [n]}$ eigenvalues and eigenvectors of $C + \epsilon\Omega_F$.

According to Theorem A.2, we have

$$\tilde{u}_j = u_j + \epsilon \sum_{k \neq j} \frac{u_k^\top \Omega_F u_j}{\lambda_j - \lambda_k} u_k + O(\epsilon^2)$$

Then,

$$\tilde{u}_j^\top X_{:i} = u_j^\top X_{:i} + \epsilon \sum_{k \neq j} \frac{u_k^\top \Omega_F u_j}{\lambda_j - \lambda_k} u_k X_{:i} + O(\epsilon^2)$$

(1) For $\{j | u_j^T X_{:i} \neq 0\}$

The leading term $u_j^T X_{:i} \neq 0$ ensures that $\tilde{u}_j^\top X_{:i} \neq 0$.

It indicates that non-zero components of $X_{:i}$ in eigenspace of $C$ is still non-zero components in eigenspace of $C + \epsilon\Omega_F$.

(2) For $\{j | u_j^T X_{:i} = 0\}$

We have

$$\tilde{u}_j^\top X_{:i} = \epsilon \sum_{k \neq j} \frac{u_k^\top \Omega_F u_j}{\lambda_j - \lambda_k} u_k X_{:i} + O(\epsilon^2)$$

$$= \epsilon \sum_{j \neq i} [\sum_{l=1}^{h} \frac{(u_k^\top X_{:l})(X_{:l}^\top u_j)}{\|X_{:l}\|_F^2 (\lambda_j - \lambda_k)}] u_k^\top X_{:i} + O(\epsilon^2)$$

When there exist $k$ that $u_k^T X_{:i} \neq 0$ and there exist $l$ that $u_k^\top X_{:l} \neq 0, u_j^\top X_{:l} \neq 0$. Thus, $(u_k^\top X_{:l})(X_{:l}^\top u_j) \neq 0$ and $\tilde{u}_k^\top X_{:i} \neq 0$.

It indicates that the zero-components of $X_{:i}$ in eigenspace of $C$ becomes non-zero components in eigenspace of $C + \epsilon\Omega_F$.

In summary, when perturbing matrix $C$ with $\epsilon\Omega_F$, the non-zero frequency component $\|\tilde{X}_{:i}^{(C+\epsilon\Omega_F)}\|_0 > \|\tilde{X}_{:i}^{(C)}\|_0$.

$\square$

**Theorem 5.4.** *For a graph $G$, the learnable matrix $\Omega(A, X)$ is commutative with $\text{Aut}(G)$ and preserves edge connectivity.*

*Proof.* (1) $\Omega(A, X)$ commutes with $Aut(G)$.

For any permutation matrix $P \in Aut(G)$, we have $PAP = A$, $P^{-1} = P^\top$ and $PDP^\top = D$.

Therefore:

$$P(D + B)P^\top = PDP^\top + PBP^\top = D + B$$
$$P(D + B)^{-1/2}P^\top = (D + B)^{-1/2}$$

For two isomorphic nodes $u \sim v$, they will have same node labels. Each element in $B$ is updated by gradient, when $u \sim v$, the gradient of $b_u$ and $b_v$ are the same. As we initial all $b_u = \frac{1}{n}$, we will get $b_u = b_v$. Thus, $PBP^\top = B$.

For $\Omega_D(A) = (D+B)^{-1/2}(A+B)(D+B)^{-1/2}$

$$P\Omega_D(A)P^\top$$
$$= P(D+B)^{-1/2}(A+B)(D+B)^{-1/2}P^\top$$
$$= P(D+B)^{-1/2}A(D+B)^{-1/2}P^\top$$
$$+ P(D+B)^{-1/2}B(D+B)^{-1/2}P^\top$$
$$= (D+B)^{-1/2}PAP^\top(D+B)^{-1/2}$$
$$+ (D+B)^{-1/2}PBP^\top(D+B)^{-1/2}$$
$$= (D+B)^{-1/2}A(D+B)^{-1/2}$$
$$+ (D+B)^{-1/2}B(D+B)^{-1/2}$$
$$= (D+B)^{-1/2}(A+B)(D+B)^{-1/2}$$
$$= \Omega_D(A)$$

Obviously, for $\Omega_S(A) = I$, we have $PIP^\top = I$, i.e., $P\Omega_S(A)P^\top = \Omega_S(A)$.

For $\Omega_F(X) = \sum_{i=1}^{h} \frac{X_{:i}X_{:i}^\top}{\|X_{:i}\|_F^2} \circ A$, we have

$$P\Omega_F(X)P^\top$$
$$= P\left(\frac{X_{:i}X_{:i}^\top}{\|X_{:i}\|_F^2} \circ A\right)P^\top$$
$$= \frac{(PX_{:i})(PX_{:i})^\top}{\|X_{:i}\|_F^2} \circ A$$
$$= \frac{X_{:i}X_{:i}^\top}{\|X_{:i}\|_F^2} \circ A$$
$$= \Omega_F(X)$$

As each term in $\Omega(A, X)$ commutes with $Aut(G)$, putting them together, we have

$$P\Omega(A, X)P^\top = \Omega(A, X)$$

(2) $\Omega(A, X)$ preserves edge connectivity.

For $\Omega_D(A) = (D+B)^{-1/2}(A+B)(D+B)^{-1/2}$, $B$ is a diagonal matrix and $A$ represents the edge connectivity, $(D+B)^{-1/2}(A+B)(D+B)^{-1/2}$ ensures that all original edges are scaled but not removed.

For $\Omega_S(D) = I$, it adds self-loops but does not affect the existing edges.

For $\Omega_F(X) = \sum_{i=1}^{h} \frac{X_{:i}X_{:i}^\top}{\|X_{:i}\|_F^2} \circ A$, the Hadamard product $\circ A$ ensures that only weights of existing edges are modified (no new edges are added), the edge connectivity is preserved.

In summary, $\Omega(A, X)$ commutes with $Aut(G)$ and preserves edge connectivity.

$\square$

**Proposition 5.5.** *When $f_W$ is permutation equivariant, spectral GNNs $\Psi^+(A, X)$ augmented with AdaSpec is permutation equivariant.*

*Proof.* The spectrum GNNs in Equation (2) has the format $\Psi^+(A, X) = g_\Theta(\Omega(A, X))f_W(X)$. We denote $M = \Omega(A, X)$ to simplify the analysis.

It has been proved in Theorem 5.4 that $M = \Omega(A, X)$ is commutative with $Aut(G)$ and preserves edge connectivity.

(1) Permuted Graph.

Let $\pi \in \text{Sym}(\mathcal{V})$ be a permutation of the nodes. Applying $\pi$ to $G$ results in a permuted graph $\pi(G)$, where both the adjacency matrix $M$ and the feature matrix $X$ are permuted:

$$\pi(M) = P_\pi M P_\pi^\top$$
$$\pi(X) = P_\pi X$$

where $P_\pi$ is the permutation matrix corresponding to $\pi$.

(2) Applying $\Psi^+$ to the Permuted Graph $\pi(G)$.

$$\Psi^+(\pi(G)) = g_\Theta(\pi(M)) f_W(\pi(X))$$
$$= \left( \sum_{k=0}^{K} \theta_k T_k(\pi(M)) \right) f_W(P_\pi X)$$

(3) Term $T_k(\pi(M))$.

Since $T_k$ is a polynomial basis and $M = \Omega(A, X)$ commutes with $P_\sigma$ for all $\sigma \in \text{Aut}(G)$, we have:

$$T_k(\pi(M)) = P_\pi T_k(M) P_\pi^\top$$

Therefore:

$$g_\Theta(\pi(M)) = \sum_{k=0}^{K} \theta_k T_k(\pi(M)) = \sum_{k=0}^{K} \theta_k P_\pi T_k(M) P_\pi^\top = P_\pi g_\Theta(M) P_\pi^\top$$

(4) Term $f_W(\pi(X))$.

As $f_W$ is permutation equivariant, we have

$$f_W(\pi(X)) = P_\pi f_W(X)$$

Therefore,

$$\Psi^+(\pi(G)) = P_\pi g_\Theta(M) P_\pi^\top \cdot P_\pi f_W(X) = P_\pi g_\Theta(M) f_W(X) = P_\pi \Psi^+(G)$$

Thus, a spectral GNN $\Psi^+(A, X)$ is permutation equivariant.

$\square$

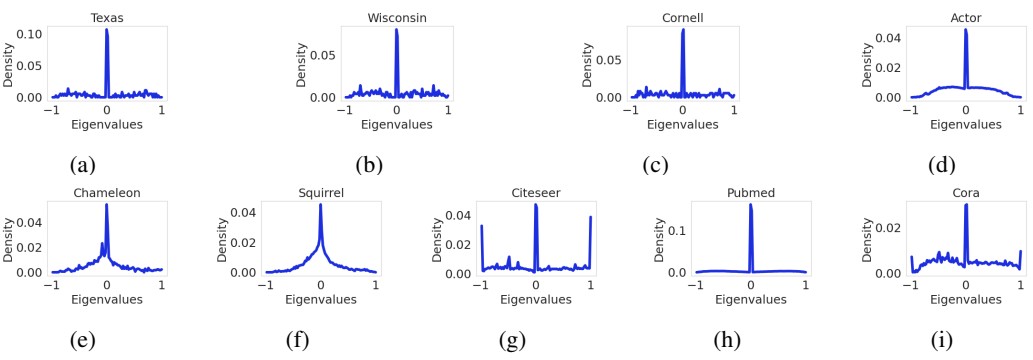

Figure 3: Distributions of eigenvalues of normalized graph adjacency matrix.

## B  EXPERIMENTAL SETTINGS AND RESULTS

We introduce statistical information of datasets, details of spectral GNNs, hyperparameter setting, distribution of graph matrix spectrum and frequency components of node features of real-world datasets and more experimental results in this section.

| Statistics | Texas | Wisconsin | Cornell | Actor | Chameleon | Squirrel |
|---|---|---|---|---|---|---|
| # Nodes | 183 | 251 | 183 | 7,600 | 890 | 2,223 |
| # Edges | 295 | 466 | 280 | 26,752 | 27,168 | 131,436 |
| # Features | 1,703 | 1,703 | 1,703 | 932 | 2,325 | 2,089 |
| # Classes | 5 | 5 | 5 | 5 | 5 | 5 |
| # Edge Homophily | 0.11 | 0.21 | 0.3 | 0.22 | 0.24 | 0.22 |

Statistics of six small heterophilic datasets (Pei et al., 2020; Rozemberczki et al., 2021; Platonov et al., 2023).

| Statistics | Roman_Empire | Amazon_Ratings | Tolokers | Minesweeper | Questions |
|---|---|---|---|---|---|
| # Nodes | 22,662 | 24,492 | 11,758 | 10,000 | 48,921 |
| # Edges | 32,927 | 93,050 | 519,000 | 39,402 | 153,540 |
| # Features | 300 | 300 | 10 | 7 | 301 |
| # Classes | 18 | 5 | 2 | 2 | 2 |
| # Edge Homophily | 0.05 | 0.38 | 0.59 | 0.68 | 0.84 |

Statistics of five large heterophilic datasets Platonov et al. (2023).

| Statistics | Citeseer | Pubmed | Cora | Computers | Photo | Coauthor-CS | Coauthor-Physics |
|---|---|---|---|---|---|---|---|
| # Nodes | 3,327 | 19,717 | 2,708 | 13,752 | 7,650 | 18,333 | 134,493 |
| # Edges | 4,676 | 44,327 | 5,278 | 491,722 | 238,162 | 163,788 | 495,924 |
| # Features | 3,703 | 500 | 1,433 | 767 | 745 | 6,805 | 8,415 |
| # Classes | 6 | 5 | 7 | 10 | 8 | 15 | 5 |
| # Edge Homophily | 0.74 | 0.8 | 0.81 | 0.78 | 0.83 | 0.81 | 0.93 |

Statistics of homophilic datasets, including three small datasets (Citeseer, Pubmed, Cora) and four large datasets (Computers, Photo, Coauthor-CS, Coauthor-Physics) (Kipf & Welling, 2017; Zeng et al., 2020; Shchur et al., 2018).

Table 7: Statistics of real-world datasets.

## B.1 DATASETS

The statistical information of the datasets, including node numbers, edge number, feature dimensions, node class numbers, edge homophilic ratios are summarized in in Table 7.

We use the directed clean version of Chameleon and Squirrel provided by (Platonov et al., 2023) which removes repeated nodes in graphs. The large heterophilic dataset is proposed in (Platonov et al., 2023). The datasets Tolokers, Minesweeper and Questions are classified as homophilic datasets under the $H_{edge}$ metric (Zhu et al., 2020), although they belong to heterophilic datasets according to the *adjusted homophily* metric in (Platonov et al., 2023).

## B.2 DATA DISTRIBUTION IN REAL-WORLD DATASETS

We show eigenvalues distributions of normalized graph adjacency matrix of real-world datasets in Figure 3. Distributions of frequency components of node feature column vectors in eigenspace of normalized graph adjacency matrix in Figure 4.

## B.3 HYPERPARAMETER SETTINGS

All experiments are run on a GPU NVIDIA RTX A6000 with 48G memory.

Following (Platonov et al., 2023), we fix the hidden size of the MLP to 512 and set early stopping with patience of 100 steps on five large heterophilic datasets (Roman_Empire, Amazon_Ratings, Tolokers, Minesweeper, Questions). Following (Chien et al., 2021; He et al., 2021), we For all other fix the hidden size of the MLP to 64 and set early stopping with patience of 200 steps on all other datatsets. The maximum number of epochs is set to 1,000.

We conduct a grid search for hyperparameters used during the training of spectral GNNs, including learning rates, dropout rates, exponential decay parameters, propagating coefficient for GPRGNN and JacobiConv, parameters $a, b$ in JacobiConv. For different datasets, we use different grid search range, The exact search ranges for different hyperparameters on different datasets are detailed in Table 8.

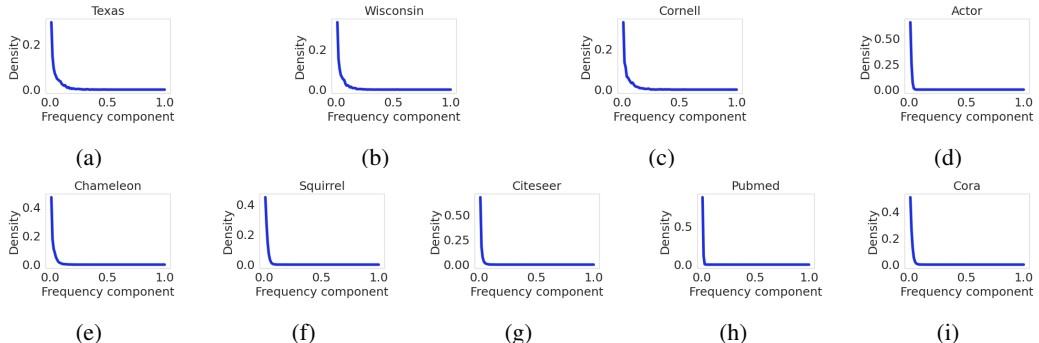

Figure 4: Distributions of frequency components of graph signals in eigenspace of normalized graph adjacency matrix.

| Datasets | Hyperparameters | GNNs | Range |
|---|---|---|---|
| 'Cora', 'Citeseer', 'Pubmed', 'Chameleon', 'Squirrel', 'Actor', 'Texas', 'Cornell', 'Wisconsin' | dropout in MLP | All/JacobiConv | $0.5, 0.7, 0.9$ |
| | dropout after MLP | All/JacobiConv | $0.5, 0.7, 0.9$ |
| | dropout in MLP | JacobiConv | $0.5, 0.7$ |
| | dropout after MLP | JacobiConv | $0.5, 0.7$ |
| | learning rate of $\Theta$ | All | $0.001, 0.01$ |
| | learning rate of $W$ | All | $0.01, 0.05$ |
| | weight decay of $\Theta$ | All | $0.0, 0.0005$ |
| | weight decay of $W$ | All | $0.0, 0.0005$ |
| | $a$ | JacobiConv | $-0.5, 0.5$ |
| | $b$ | JacobiConv | $-0.5, 0.5$ |
| | propagation parameter $\alpha$ | JacobiConv | $0.1, 0.9$ |
| | propagation parameter $\alpha$ | GPRGNN | $0.1, 0.2, 0.9$ |
| 'amazon_ratings', 'minesweeper', 'questions', 'roman_empire', 'tolokers' | dropout in MLP | All | $0.5$ |
| | dropout after MLP | All | $0.5, 0.7$ |
| | learning rate of $\Theta$ | All | $0.001, 0.01$ |
| | learning rate of $W$ | All | $0.01, 0.05$ |
| | weight decay of $\Theta$ | All | $0.0, 0.0005$ |
| | weight decay of $W$ | All | $0.0, 0.0005$ |
| | $a$ | JacobiConv | $-0.5, 0.5$ |
| | $b$ | JacobiConv | $-0.5, 0.5$ |
| | propagation parameter $\alpha$ | JacobiConv | $0.1, 1.0$ |
| | propagation parameter $\alpha$ | GPRGNN | $0.0, 0.9$ |
| 'computers', 'photo', 'coauthor-cs', 'coauthor-physics' | dropout in MLP | All | $0.5, 0.7$ |
| | dropout after MLP | All | $0.5, 0.7$ |
| | learning rate of $\Theta$ | All | $0.001, 0.01$ |
| | learning rate of $W$ | All | $0.01, 0.05$ |
| | weight decay of $\Theta$ | All | $0.0, 0.0005$ |
| | weight decay of $W$ | All | $0.0, 0.0005$ |
| | $a$ | JacobiConv | $-0.5, 0.5$ |
| | $b$ | JacobiConv | $-0.5, 0.5$ |
| | propagation parameter $\alpha$ | JacobiConv | $0.1, 0.9$ |
| | propagation parameter $\alpha$ | GPRGNN | $0.1, 0.2, 0.9$ |

Table 8: Grid search ranges of hyperparameters. Dropout search ranges of JacobiConv is smaller than other spectral GNNs as it contains too many hyperparameters, we have to reduce the search range to guarantee that the searching process can be finished in accepted computing time.

### B.4 SPECTRAL GNNs

We provide the detailed description for spectral GNNs used in our experiments in the following.

For a graph with the adjacency matrix $A$, the degree matrix $D$, and the identity matrix $I$, we use $\hat{L} = I - D^{-1/2}AD^{-1/2}$, $\tilde{L} = -D^{-1/2}AD^{-1/2}$, $\tilde{A} = D^{-1/2}AD^{-1/2}$, and $\tilde{A}' = (D+I)^{-1/2}(A+I)(D+I)^{-1/2}$ to denote the normalized Laplacian matrix, the shifted normalized Laplacian matrix, the normalized adjacency, matrix and the normalized adjacency matrix with self-loops, respectively.

**ChebNet** (Defferrard et al., 2016): This model uses the Chebyshev basis to approximate a spectral filter:

$$\hat{Y} = \sum_{k=0}^{K} \theta_k T_k(\tilde{L}) f_W(X)$$

where $X$ is the raw feature matrix, $\Theta = [\theta_0, \theta_1, \dots, \theta_K]$ is the graph convolution parameter, $W$ is the feature transformation parameter and $f_W(X)$ is usually a 2-layer MLP. $T_k(\tilde{L})$ is the $k$-th Chebyshev basis expanded on the shifted normalized graph Laplacian matrix $\tilde{L}$ and is recursively calculated:

$$T_0(\tilde{L}) = I$$
$$T_1(\tilde{L}) = \tilde{L}$$
$$T_k(\tilde{L}) = 2\tilde{L}T_{k-1}(\tilde{L}) - T_{k-2}(\tilde{L})$$

**ChebNetII** (He et al., 2022a): The model is formulated as

$$\hat{Y} = \frac{2}{K+2} \sum_{k=0}^{K} \sum_{j=0}^{K} \theta_j T_k(x_j) T_k(\tilde{L}) f_W(X),$$

where $X$ is the input feature matrix, $W$ is the feature transformation parameter, $f_W(X)$ is usually a 2-layer MLP, $T_k(\cdot)$ is the $k$-th Chebyshev basis expanded on $\cdot$, $x_j = \cos\left((j + 1/2)\pi / (K + 1)\right)$ is the $j$-th Chebyshev node, which is the root of the Chebyshev polynomials of the first kind with degree $K + 1$, and $\theta_j$ is a learnable parameter. Graph convolution parameter in ChebNet is reparameterized with Chebyshev nodes and learnable parameters $\theta_j$.

**JacobiNet** (Wang & Zhang, 2022): This model uses the Jacobi basis to approximate a filter as:

$$\hat{Y} = \sum_{k=0}^{K} \theta_k P_k^{a,b}(\tilde{A}) f_W(X),$$

where $X$ is the input feature matrix, $\Theta = [\theta_0, \theta_1, \dots, \theta_K]$ is the graph convolution parameter, $W$ is the feature transformation parameter and $f_W(X)$ is usually a 2-layer MLP. $P_k^{a,b}(\tilde{A})$ is the Jacobi basis on normalized graph adjacency matrix $\tilde{A}$ and is recursively calculated as

$$P_k^{a,b}(\tilde{A}) = I$$
$$P_k^{a,b}(\tilde{A}) = \frac{1-b}{2}I + \frac{a+b+2}{2}\tilde{A}$$
$$P_k^{a,b}(\tilde{A}) = \gamma_k \tilde{A} P_{k-1}^{a,b}(\tilde{A}) + \gamma_k' P_{k-1}^{a,b}(\tilde{A}) + \gamma_k'' P_{k-2}^{a,b}(\tilde{A})$$

where $\gamma_k = \frac{(2k+a+b)(2k+a+b-1)}{2k(k+a+b)}, \gamma_k' = \frac{(2k+a+b-1)(a^2-b^2)}{2k(k+a+b)(2k+a+b-2)}, \gamma_k'' = \frac{(k+1-1)(k+b-1)(2k+a+b)}{k(k+a+b)(2k+a+b-2)}$. $a$ and $b$ are hyperparameters. Usually, grid search is used to find the optimal $a$ and $b$ values.

**GPRGNN** (Chien et al., 2021): This model uses the monomial basis to approximate a filter:

$$\hat{Y} = \sum_{k=0}^{K} \theta_k \tilde{A}'^k f_W(X)$$

where $X$ is the input feature matrix, $\Theta = [\theta_0, \theta_1, \ldots, \theta_K]$ is the graph convolution parameter, $W$ is the feature transformation parameter and $f_W(X)$ is usually a 2-layer MLP. $\tilde{A}'$ is the normalized adjacency matrix with self-loops.

**BernNet** (He et al., 2021): This model uses the Bernstein basis for approximation:

$$\hat{Y} = \sum_{k=0}^{K} \theta_k \frac{1}{2^K} \binom{K}{k} (2I - \hat{L})^{K-k} \hat{L}^k f_W(X)$$

where $X$ is the input feature matrix, $\Theta = [\theta_0, \theta_1, \ldots, \theta_K]$ is the graph convolution parameter, $W$ is the feature transformation parameter and $f_W(X)$ is usually a 2-layer MLP. $\hat{L}$ is the normalized Laplacian matrix.

### B.5 FULL EXPERIMENTAL RESULTS ON LARGE HETEROPHILIC GRAPHS

We show our full experimental results on large heterophilic graphs in Table 9. There is an average 1.08% accuracy improvement on Roman_Empire, Amazon_Ratings and an average 1.1% ROC AUC improvement on the rest datasets.

| Model | Roman_Empire | Amazon_Ratings | Tolokers | Minesweeper | Questions |
|---|---|---|---|---|---|
| ChebNet(O) | $47.15_{\pm 0.42}$ | $39.79_{\pm 0.29}$ | $70.1_{\pm 0.25}$ | $86.29_{\pm 0.2}$ | $55.13_{\pm 0.54}$ |
| cheb (M) | $54.55_{\pm 0.3}$ | $40.92_{\pm 0.27}$ | $69.2_{\pm 0.61}$ | $86.7_{\pm 0.23}$ | $55.2_{\pm 1.52}$ |
| $\Delta \uparrow$ | +7.4 | +1.13 | -0.9 | +0.41 | +0.07 |
| ChebNetII (O) | $55.44_{\pm 0.19}$ | $39.99_{\pm 0.28}$ | $69.93_{\pm 0.83}$ | $78.35_{\pm 0.14}$ | $64.13_{\pm 0.95}$ |
| ChebNetII (M) | $55.1_{\pm 0.35}$ | $40.66_{\pm 0.33}$ | $70.94_{\pm 0.36}$ | $79.1_{\pm 0.09}$ | $65.54_{\pm 0.7}$ |
| $\Delta \uparrow$ | -0.34 | +0.67 | +1.01 | +0.75 | +1.41 |
| JacobiConv (O) | $55.86_{\pm 0.57}$ | $40.27_{\pm 0.3}$ | $70.1_{\pm 0.22}$ | $87.34_{\pm 0.12}$ | $64.72_{\pm 0.38}$ |
| JacobiConv (M) | $56.21_{\pm 0.38}$ | $40.17_{\pm 0.24}$ | $71.04_{\pm 0.22}$ | $89.13_{\pm 0.1}$ | $65.8_{\pm 0.18}$ |
| $\Delta \uparrow$ | +0.35 | -0.1 | +0.94 | +1.79 | +1.08 |
| GPRGNN (O) | $56.33_{\pm 1.51}$ | $40.07_{\pm 0.25}$ | $66.34_{\pm 1.76}$ | $87.15_{\pm 0.49}$ | $53.14_{\pm 0.27}$ |
| GPRGNN (M) | $56.96_{\pm 1.59}$ | $40.14_{\pm 0.38}$ | $68.44_{\pm 0.39}$ | $88.58_{\pm 0.18}$ | $58.19_{\pm 0.36}$ |
| $\Delta \uparrow$ | +0.63 | +0.07 | +2.1 | +1.43 | +5.05 |
| BernNet (O) | $55.06_{\pm 0.3}$ | $39.36_{\pm 0.37}$ | $68.81_{\pm 0.91}$ | $76.54_{\pm 0.23}$ | $64.86_{\pm 0.37}$ |
| BernNet (M) | $55.51_{\pm 0.91}$ | $39.85_{\pm 0.23}$ | $69.49_{\pm 0.72}$ | $76.95_{\pm 0.21}$ | $65.2_{\pm 0.31}$ |
| $\Delta \uparrow$ | +0.45 | +0.49 | +0.68 | +0.41 | +0.34 |

Table 9: Performance with/without AdaSpec on large heterophilic datasets (Roman_Empire, Amazon_Ratings, Tolokers, Minesweeper, Questions ). Test accuracy is used as the metric for Roman-Empire and Amazon-Ratings datasets and ROC AUC is reported on Minesweeper, Tolokers, Questions. High accuracy and ROC AUC indicate good performance.

| Model | Texas | Wisconsin | Actor | Chameleon | Squirrel | Cornell | Citeseer | Pubmed | Cora |
|---|---|---|---|---|---|---|---|---|---|
| ChebNet(O) | $38.67_{\pm 9.31}$ | $32.92_{\pm 7.38}$ | $25.15_{\pm 0.69}$ | $29.32_{\pm 4.13}$ | $24.23_{\pm 3.24}$ | $31.33_{\pm 7.51}$ | $69.21_{\pm 0.87}$ | $75.29_{\pm 2.34}$ | $80.45_{\pm 1.09}$ |
| GDC + ChebNet(O) | $50.58_{\pm 8.13}$ | $34.00_{\pm 7.62}$ | $24.92_{\pm 0.73}$ | $21.52_{\pm 2.62}$ | $20.62_{\pm 1.57}$ | $28.50_{\pm 7.63}$ | $67.52_{\pm 1.37}$ | $74.53_{\pm 1.95}$ | $75.91_{\pm 1.36}$ |
| **GDC + ChebNet(M)** | $\mathbf{52.14}_{\pm 8.27}$ | $\mathbf{36.33}_{\pm 9.38}$ | $24.15_{\pm 1.02}$ | $\mathbf{31.84}_{\pm 2.68}$ | $\mathbf{22.02}_{\pm 4.59}$ | $\mathbf{34.91}_{\pm 10.8}$ | $\mathbf{68.26}_{\pm 0.98}$ | $\mathbf{77.22}_{\pm 1.43}$ | $\mathbf{81.01}_{\pm 1.11}$ |
| $\Delta \uparrow$ | +1.56 | +2.33 | -0.77 | +10.32 | +1.40 | +6.41 | +0.74 | +2.69 | +5.10 |

Table 10: Impact of AdaSpec applied on top of GDC. Our method (GDC + ChebNet(M)) consistently improves performance across most benchmarks. Three configurations: (1) Standard ChebNet ( ChebNet(O) ), (2) ChebNet with GDC ( GDC+ChebNet(O) ), and (3) ChebNet with GDC + AdaSpec ( GDC+ChebNet(M) ).

### B.6 EXPERIMENTAL RESULTS OF ADASPEC WITH GRAPH DIFFUSION CONVOLUTION

Our AdaSpec is not a competitor to graph rewiring; it is a plug-and-play spectral enhancement. We demonstrate below that, it can be seamlessly integrated with existing graph rewiring methods like graph diffusion convolution (GDC) to achieve superior performance, validating its unique value proposition beyond standard rewiring.

we conducted a set of experiments combining AdaSpec with GDC and results are shown in Table 10. Our key findings are as follows. (1) Performance improvement: GDC+ChebNet(M) improves performance than GDC+ChebNet(O) In 8 out of 9 reported cases. (2) Orthogonality: If AdaSpec and GDC were solving the exact same problem (competitive mechanisms), stacking them would yield diminishing returns. The fact that AdaSpec provides significant gains on top of GDC proves they address orthogonal limitations of the graph structure. (3) Spatial rewiring optimizes topological connectivity (e.g., denoising edges to improve homophily). AdaSpec optimizes the eigenspace. Even a graph that is topologically clean (via GDC) may still suffer from eigenvalue multiplicity or missing frequency components in the spectral domain. AdaSpec resolves these spectral collisions, which GDC cannot detect or repair.

