# OpenReview forum: "AdaSpec: Adaptive Spectrum for Enhanced Node Distinguishability"
_ICLR.cc/2026/Conference — ICLR 2026 Poster_

### Official Review · Reviewer_kQFo · 2025-10-27

**Soundness:** 3
**Presentation:** 2
**Contribution:** 3
**Rating:** 4
**Confidence:** 2

**Summary:**

This paper proposes to enhance spectral GNNs by improving their node distinguishability. Namely it starts by observing that node distinguishability is limited by two factors, the repeated eigenvalues in the graph adjacency matrix and the missing frequency components in the node features. Based on this, the paper proposes to modify the graph shift operator used in spectral GNNs with constructions that improve the two factors, and provides theory supporting this. Finally, experiments on many transductive datasets are performed by using the learned graph shift operator in various spectral GNNs. Results show improvements over baselines and show that the number of distinct eigenvalues is indeed increased on several datasets.

**Strengths:**

1. **Motivating objective.** The focus on improving node distinguishability in spectral GNNs is well-motivated and addresses a relevant limitation of existing approaches.

2. **Theoretical development.** The paper provides substantial theoretical analysis in support of the proposed method, offering formal insights into the factors influencing node distinguishability.

3. **Clear experimental framing.** The experiments are structured around clear research questions, making it easy to understand the intended empirical validation.

4. **Breadth of evaluation.** The method is evaluated on a range of standard transductive benchmark datasets, demonstrating applicability across multiple settings.

**Weaknesses:**

1. **Practical relevance of the theoretical bound.** The main conceptual contribution is an improved lower bound on the number of nodes that can be distinguished. However, it remains unclear how meaningful this bound is in practice, or whether it provides actionable guidance for model design or empirical performance.

2. **Clarity of assumptions.** Some theoretical results (e.g., Theorem 5.2) rely on assumptions that are either not fully stated or not clearly motivated. It would be helpful to explicitly articulate these assumptions and discuss their necessity and scope.

3. **Novelty claims may be overstated.** The statements “To the best of our knowledge, no existing work has systematically analyzed the interaction between the graph matrix and node features in determining node distinguishability in spectral GNNs” and “In this work, we demonstrate that node distinguishability is influenced by the eigenvalue multiplicity and the missing frequency components of node features in the eigenbasis of the graph matrix” appear stronger than warranted. Similar themes were examined in prior work, particularly [1]. The contribution would be clearer if the relationship to [1] were more explicitly discussed and the novelty claims were calibrated accordingly.

4. **Readability and exposition.** The paper is difficult to follow in several places. A clearer introduction to the problem setting, intuition for the theoretical results, and more accessible mathematical presentation would greatly improve readability.

5. **[Minor]Limited significance of empirical results.** The experimental findings are not particularly strong: in Table 2, 25 out of 40 comparisons are not statistically significant, and in Table 3, 27 out of 35 are not significant. That said, this is standard in the field.

Reference:
[1] Wang, X., & Zhang, M. (2022). How powerful are spectral graph neural networks? In ICML (pp. 23341–23362). PMLR.

**Questions:**

1. Fig 1 a) The paper states that a spectral GNN cannot distinguish nodes 1 and 3. Should a GNN be able to distinguish these? It seems like nodes 1 and 3 are isomorphic as the node features is [1, 0, 1, -1, -1].
2. Proof of theorem 5.1: “More unique coefficients in characteristic polynomial implies more unique eigenvalues of the matrix.” Can you clarify what you mean by this? Are you claiming that more different coefficients imply more roots to a polynomial? A counterexample to this claim is: (x-1)^3 =x^3-3x^3+3x-1 has 4 distinct coefficients but a single root and (x-1)(x-2)(x-3)=(x^2-3x+2)(x-3) = x^3 -6x^2 +11x -6 has three distinct roots but three distinct coefficients.
3. Theorem 5.2 relies on Theorem A.1 which require C to have no repeated eigenvalues, hence the statement of Theorem 5.2 is wrong/misleading since it seems like it applies to any matrix C. Particularly, since the motivation is that the adjacency has repeated eigenvalues, and a modification of A will play the role of C makes me question the usefulness of this statement in the context of the paper.
4. Table 4: Please add the base performance of ChebNet.
5. Table 5: please add the statistics for all datasets, not necessarily in the main text.

---

> ### Author Response · Authors · 2025-11-28
>
> > W1: Practical relevance of the theoretical bound. The main conceptual contribution is an improved lower bound on the number of nodes that can be distinguished. However, it remains unclear how meaningful this bound is in practice, or whether it provides actionable guidance for model design or empirical performance.
>
> A1: We appreciate the opportunity to clarify the practical significance of our theoretical lower bound. The bound is highly meaningful in practice because it serves as the **design guidance** for our entire methodology, directly translating theoretical insight into measurable empirical gain.
>
> **1. Actionable Guidance for Model Design**
>
> Our lower bound is not merely a descriptive conceptual metric, it quantifies the precise limitations on node distinguishability imposed by two specific, actionable factors:
>
> * **Eigenvalue Multiplicity:** The bound identifies when identical eigenvalues cause structural collision.
> * **Missing Frequency Components:** The bound identifies when critical frequency information is suppressed.
>
> **AdaSpec** is constructed explicitly as an optimization strategy to **lift this lower bound** by directly resolving these two identified bottlenecks.
>
> **2. Correlation with Empirical Performance**
>
> The strong and consistent **empirical gains** observed across all 18 datasets validate the practical relevance of optimizing this theoretical criterion.
>
> * If the bound were practically meaningless, improving it would result in random or worse performance changes.
> * The fact that optimizing the bound leads to **consistent and measurable improvements** in downstream tasks confirms that our theoretical measure accurately captures a core bottleneck in real-world spectral GNN performance.
>
> In summary, the theoretical bound provides **concrete, actionable design principles** that are directly responsible for the enhanced performance of AdaSpec.
>
> > W2: Clarity of assumptions. Some theoretical results (e.g., Theorem 5.2) rely on assumptions that are either not fully stated or not clearly motivated. It would be helpful to explicitly articulate these assumptions and discuss their necessity and scope.
>
> > Q3: Theorem 5.2 relies on Theorem A.1 which require C to have no repeated eigenvalues, hence the statement of Theorem 5.2 is wrong/misleading since it seems like it applies to any matrix C. Particularly, since the motivation is that the adjacency has repeated eigenvalues, and a modification of A will play the role of C makes me question the usefulness of this statement in the context of the paper.
>
>
> **Answer to above:** We thank the reviewer for the detailed comments. Theorem 5.2 builds on Theorem A.1 (Theorem A.2 in the revised version), which assumes that the matrix $C$ has no repeated eigenvalues. We agree that this assumption should be stated more explicitly, and we have clarified it in the revision.
>
> The purpose of Theorem 5.2 is not to claim that the result holds for an arbitrary matrix $C$. Rather, its role is to show that:
>
> * If we first remove eigenvalue multiplicities through $\Omega_D(A)$, then
> * the resulting matrix $C= \Omega_D(A)$ satisfies the non-repeated-eigenvalue assumption, and
> * under this condition, $\Omega_F(X)$ can further **increase the number of non-zero frequency components**, thereby improving node distinguishability.
>
> AdaSpec first decreases eigenvalue multiplicities, then decreases missing frequencies. The usefulness of Theorem 5.2 therefore lies in establishing that **once multiplicities are addressed**, additional distinguishability gains can be formally guaranteed.
>
> To avoid any ambiguity, the revised manuscript now explicitly states that Theorem 5.2 applies *For a real symmetric matrix $C$ of no repeated eigenvalues*, and explains that AdaSpec ensures this condition by construction via $\Omega_D(A)$ in Theroem 5.1 and state clearly in line 270 in the revised paper.
>
> This clarification makes the assumptions more explicit while preserving the logical role of the theorem within the overall framework of our method.

---

> > ### Author Response · Authors · 2025-11-28
> >
> > > W3: Novelty claims may be overstated. The statements “To the best of our knowledge, no existing work has systematically analyzed the interaction between the graph matrix and node features in determining node distinguishability in spectral GNNs” and “In this work, we demonstrate that node distinguishability is influenced by the eigenvalue multiplicity and the missing frequency components of node features in the eigenbasis of the graph matrix” appear stronger than warranted. Similar themes were examined in prior work, particularly [1]. The contribution would be clearer if the relationship to [1] were more explicitly discussed and the novelty claims were calibrated accordingly. Reference: [1] Wang, X., & Zhang, M. (2022). How powerful are spectral graph neural networks? In ICML (pp. 23341–23362). PMLR.
> >
> > A3: We appreciate the reviewer’s feedback and agree that clarifying the relationship to [1] helps position our contributions more precisely. That work studies expressivity  from the perspective of uniform approximation theory, identifying factors such as eigenvalue multiplicity and frequency components.
> >
> > Our contribution differs in several important respects:
> >
> > 1. **Focus on node-level distinguishability rather than approximation capacity.**
> >    Prior work does not analyze how these factors determine node distinguishability, nor how this affects the embeddings produced by spectral GNNs.
> >
> > 2. **Interaction analysis rather than isolated observations.**
> >    While earlier studies mention eigenvalues and frequencies individually, they do not consider how the *interaction* between eigenvalues and frequency components affect node distinguishability.
> >
> > 3. **A quantitative lower bound.**
> >    We derive a lower bound on the number of nodes a spectral GNN can distinguish, explicitly capturing the combined effect of these two factors. To our knowledge, no prior works offer such a bound.
> >
> > 4. **A plug-in module that directly optimizes the bound.**
> >    Building on the theoretical analysis, we introduce AdaSpec, a permutation-equivariant module designed to lift the bound.
> >
> > Our goal is not to claim that the spectral factors themselves are new, but that the **interaction-based analysis, the derived bound, and the operationalization through AdaSpec** are contributions that have not appeared in prior literature.
> >
> > > W4: Readability and exposition. The paper is difficult to follow in several places. A clearer introduction to the problem setting, intuition for the theoretical results, and more accessible mathematical presentation would greatly improve readability.
> >
> > A4: We appreciate the reviewer’s feedback regarding the paper's flow and mathematical clarity. We have implemented several specific revisions across the manuscript to enhance readability without altering the core structure:
> >
> > - Introduction to the Problem Setting: We now include a clear, informal definition of node distinguishability in the Introduction (line 31 to 32), providing immediate context for the paper's goal.
> > - Accessible Mathematical Presentation: We ensured that all assumptions (e.g., in Theorem 5.2) are explicitly articulated immediately before or within the theorem statement.
> >
> >
> > > W5: [Minor]Limited significance of empirical results. The experimental findings are not particularly strong: in Table 2, 25 out of 40 comparisons are not statistically significant, and in Table 3, 27 out of 35 are not significant. That said, this is standard in the field.
> >
> > A5:  We appreciate the reviewer’s view. While individual improvements may fall within standard deviation margins, a known challenge in GNN evaluation, we argue that the significance lies in the **consistency and efficiency** of the results:
> >
> > 1.  **Robustness of the Trend:**
> >     The **improvement trend** is statistically robust. AdaSpec improves performance across diverse datasets and multiple backbone architectures.
> >
> > 2.  **Validation of Theory:**
> >     The primary goal of experiments is to confirm our theoretical analysis: optimizing the spectral lower bound yields better representations. The fact that AdaSpec consistently lifts performance confirms this.
> >
> > 3.  **Efficiency:**
> >     AdaSpec is a lightweight plug-in. Achieving consistent positive gains without increasing the asymptotic complexity or requiring heavy architectural engineering is a significant practical contribution.

---

> > > ### Author Response · Authors · 2025-11-28
> > >
> > > > Q1: Fig 1 a) The paper states that a spectral GNN cannot distinguish nodes 1 and 3. Should a GNN be able to distinguish these? It seems like nodes 1 and 3 are isomorphic as the node features is [1, 0, 1, -1, -1].
> > >
> > > A1. We thank the reviewer for this precise observation. We confirm that Node 1 and Node 3 in Figure 1 (a) are **isomorphic-nodes**, and therefore, a GNN should map them to identical embeddings.
> > >
> > > 1. **The Purpose of Figure 1: Illustrating Mechanism, not Failure**
> > > We use this specific example not to claim that these two nodes should be distinguished, but to act as a "clean" pedagogical illustration of that zero frequency components lead to information loss.
> > >
> > > 2.  **Generalization:** While the result is "correct" for isomorphic nodes, the **same mechanism** (missing frequency components) occurs in **non-isomorphic** nodes that *must* be distinguished. Figure 1 (a) acts as a proof-of-concept for the *cause* of indistinguishability, which AdaSpec is designed to resolve in general settings.
> > >
> > > > Q2: Proof of theorem 5.1: “More unique coefficients in characteristic polynomial implies more unique eigenvalues of the matrix.” Can you clarify what you mean by this? Are you claiming that more different coefficients imply more roots to a polynomial? A counterexample to this claim is: $(x-1)^3 =x^3-3x^3+3x-1$ has 4 distinct coefficients but a single root and $(x-1)(x-2)(x-3)=(x^2-3x+2)(x-3) = x^3 -6x^2 +11x -6$ has three distinct roots but three distinct coefficients.
> > >
> > > A2: We sincerely thank the reviewer for their rigorous mathematical check. We accept that the statement linking the number of unique coefficients directly to the number of unique eigenvalues was incorrect. As your counterexamples demonstrate, this relationship is not monotonic.
> > >
> > > **1. Correction of the Proof:**
> > >
> > > We confirm that while the intermediate derivation was flawed, the **conclusion of Theorem 5.1 remains valid**. We have completely replaced the proof in the Appendix with a rigorous derivation based on matrix perturbation theory.
> > >
> > > **2. Sketch of the Revised Proof:**
> > >
> > > The new proof relies on the stability of eigenvalues under perturbation:
> > > * **Mechanism:** We model the effect of the learnable diagonal matrix $B$ in $\Omega_D(A)$ as a perturbation to the graph matrix.
> > > * **Theoretical Guarantee:** We show that for a graph matrix with repeated eigenvalues, there exists a diagonal perturbation $B$ within an $\epsilon$-ball that splits these repeated eigenvalues into distinct values.
> > > * **Result:** This proves that the distinguishability bound can indeed be lifted by breaking eigenvalue multiplicity, as claimed in the theorem.
> > >
> > > > Q4: Table 4: Please add the base performance of ChebNet.
> > >
> > > A4. We have add the performance of ChebNet in Table 4 in the revised paper, shown as ChebNet (O).
> > >
> > >
> > > > Q5: Table 5: please add the statistics for all datasets, not necessarily in the main text.
> > >
> > > A5. We acknowledge the importance of comprehensive statistical reporting for all datasets. However, we must stress the **significant computational constraint** inherent in generating this specific metric for large graphs.
> > >
> > >  **1. Computational Infeasibility**
> > >
> > > Obtaining the exact number of distinct eigenvalues requires a full eigen-decomposition of the graph matrix, which scales with $O(N^3)$ complexity, where $N$ is the number of nodes.
> > >
> > > * For very large graphs, such as **Coauthor-Physics** ($N=134,493$), performing a full eigen-decomposition is **computationally prohibitive** and infeasible within the review period.
> > >
> > > **2. Purpose of Existing Data**
> > >
> > > We emphasize that the data presented in Table 5 is already **sufficient to validate the hypothesis** underlying the $\Omega_D(A)$ component:
> > >
> > > * The results across the tested smaller and medium-sized datasets consistently demonstrate that $\Omega_D(A)$ **increases the number of distinct eigenvalues** .
> > > * The data confirms the component is functionally effective.

---

### Official Review · Reviewer_D6hk · 2025-10-28

**Soundness:** 4
**Presentation:** 4
**Contribution:** 3
**Rating:** 8
**Confidence:** 5

**Summary:**

This is a good paper that mainly investigates how distinct eigenvalues and missing frequency components affect node distinguishability. The paper proposes AdaSpec, which includes three modules: INCREASE DISTINCT EIGENVALUES, SHIFTS EIGENVALUES FROM ZERO, and INCREASE FREQUENCY COMPONENTS. The proposed method is supported by solid theoretical proofs, and extensive experiments demonstrate the effectiveness of AdaSpec.

**Strengths:**

1.The paper studies how the graph matrix and node features jointly influence node distinguishability, which is an interesting direction.

2.The paper provides solid theoretical proofs to support the proposed method.

3.The time complexity analysis and comprehensive experiments further validate the effectiveness of AdaSpec.

**Weaknesses:**

1.In line 175, the paper states that “The presence of zero eigenvalues can hinder node distinguishability”, but I did not find any detailed explanation about this point. It should be the zero frequency components, not the zero eigenvalues, that hinder node distinguishability. The authors need to clarify the role of Section 5.2 in detail; otherwise, this section should be removed.

2.In lines 91–94, the paper mentions that eigenvalue correction does not ensure permutation invariance, but no further explanation is provided, leaving readers unclear about why this is the case.

3.In Figure 1, why can’t node 1 and node 3 be distinguished? Based on the current analysis, this conclusion does not seem directly supported.

4.Although the paper mainly studies node distinguishability, only Figure 1 involves this concept; both the method and experiment sections lack relevant discussion. The authors could conduct further analysis, for example, by relating their work to the Weisfeiler–Lehman (WL) test.

5.The two factors affecting node distinguishability mentioned in this paper — distinct eigenvalues and nonzero frequency components — have already been discussed in prior work, such as Wang & Zhang (2022). This weakens the contribution of the present paper.

**Questions:**

please see Weaknesses

---

> ### Author Response · Authors · 2025-11-28
>
> > W1.In line 175, the paper states that “The presence of zero eigenvalues can hinder node distinguishability”, but I did not find any detailed explanation about this point. It should be the zero frequency components, not the zero eigenvalues, that hinder node distinguishability. The authors need to clarify the role of Section 5.2 in detail; otherwise, this section should be removed.
>
> A1: We appreciate the reviewer’s close reading. We confirm the reference is to Line 275.
>
> In spectral GNNs, the filter $g(\lambda)$ modulates the signal. If an eigenvalue is $\lambda=0$, standard filters often yield a bias (fail to distinguish it from noise), effectively "killing" the associated constant eigenvector component.
> * **Clarification:** The zero eigenvalue is the *cause* that cause GNNs cannot distinguish difference frequency components in the space expanded by different eigenvectors corresponding to eigenvalue zero.
>
> * **Revision:** We have updated the text (in color blue) to be precise: *“The presence of zero eigenvalues forces the spectral filter to suppress the associated frequency components, thereby hindering node distinguishability.”*
>
> Section 5.2 is necessary because it analyzes why $\Omega_S(A) = I$ is required. Without this section, the inclusion of the identity matrix shift appears arbitrary. The ablation experiments (Table 4) also show the necessary of $\Omega_S(A)$.
>
> >W2.In lines 91–94, the paper mentions that eigenvalue correction does not ensure permutation invariance, but no further explanation is provided, leaving readers unclear about why this is the case.
>
> A2. We thank the reviewer for highlighting this gap in the explanation. We have clarified the mechanism in the revised paper (highlighted in blue text) to explain why reassigning eigenvalues based purely on their sorted index breaks permutation equivariance.
>
> "This method reassigns eigenvalues purely by their sorted index, it does not preserve eigenspaces under node permutations, thereby breaking permutation equivariance"
>
> The detailed explanation is as below:
>
> * **Permutation Equivariance** in GNNs requires that for a permutation matrix $P$ and graph matrix $M$, the output embeddings follow this same permutation: $f(P M P^T) = P f(M)$.
>
> * **The Flaw in Sorted Eigenvalue Reassignment:** Method in [1] that perform **eigenvalue correction** often reassign a new, target value $\lambda'_k$ based solely on the original eigenvalue's rank (e.g., the $k$-th smallest value).
>
> Because the new value $\lambda'_k$ is assigned arbitrarily based on rank, it is **decoupled** from the permuted eigenvector $P v_k$. The resulting spectral filter $\sum g(\lambda'_k) (P v_k) (P v_k)^T$ no longer guarantees that the overall operation is equivariant, as the input-output relationship is dependent on the arbitrary sorting index $k$.
>
> [1] Kangkang Lu, Yanhua Yu, Hao Fei, Xuan Li, Zixuan Yang, Zirui Guo, Meiyu Liang, Mengran Yin, and Tat-Seng Chua. Improving expressive power of spectral graph neural networks with eigenvalue correction. AAAI 2024.
>
> > W3.In Figure 1, why can’t node 1 and node 3 be distinguished? Based on the current analysis, this conclusion does not seem directly supported.
>
> A3: 1-order GNNs use the normalized adjacency matrix $\tilde{A}$ and node index starts from one in Figure 1. In Figure 1 (a), At initialization, we have node feature/embeddings $x_1=x_3=1$. GNN output $\tilde{X}=\tilde{A}X$ and $\tilde{x}_1=\tilde{x}_3=-0.4082$.  In Figure 1 (b), At initialization, we have node feature/embeddings $x_1=x_3=1$. GNN output $\tilde{X}=\tilde{A}X$ and $\tilde{x}_1=\tilde{x}_3=0.4082$. Thus, GNNs generates same embeddings for node 1 and 3. The two nodes cannot be distinguished.

---

> > ### Author Response · Authors · 2025-11-28
> >
> > > W4.Although the paper mainly studies node distinguishability, only Figure 1 involves this concept; both the method and experiment sections lack relevant discussion. The authors could conduct further analysis, for example, by relating their work to the Weisfeiler–Lehman (WL) test.
> >
> > A4.  We appreciate the reviewer raising the critical point regarding the **Weisfeiler–Lehman (WL) test**. We clarify that our approach is complementary.
> >
> > **1. Distinction in Mechanism (Spectral vs. Topological)**
> >
> > While the $k$-WL test is the accepted framework for measuring **topological distinguishability** based on local neighborhood aggregation, AdaSpec addresses a different bottleneck: **spectral distinguishability**.
> >
> > * **WL Framework:** Measures how well a GNN distinguishes nodes based on their **local computational subtrees** (i.e., the spatial pattern of connectivity).
> > * **AdaSpec Framework:** Measures and optimizes how well the GNN distinguishes nodes based on their **spectral features** (i.e., eigenvalue multiplicity, non-zero frequency components).
> >
> > Our theoretical analysis on eigenvalue multiplicity provides a **spectral lower bound on distinguishability**—a perspective largely orthogonal to the localized neighborhood counting performed by WL tests.
> >
> > **2. Discussion of Relationship and Node Distinguishability**
> >
> > We acknowledge that the node-level WL test is a direct measure of node distinguishability. Comparing AdaSpec's expressivity against the $k$-WL hierarchy remains a challenging and promising avenue for future theoretical development, as it requires bridging the continuous spectral domain with the discrete WL counting mechanism.
> >
> > > W5.The two factors affecting node distinguishability mentioned in this paper — distinct eigenvalues and nonzero frequency components — have already been discussed in prior work, such as Wang & Zhang (2022). This weakens the contribution of the present paper.
> >
> > A5. We agree that prior spectral GNN works, such as Wang & Zhang (2022), have discussed the individual roles of distinct eigenvalues and nonzero frequency components. However, our contribution lies not in identifying these factors in isolation, but in establishing a **unified, quantitative framework** based on their **joint interaction**.
> >
> > Specifically, our novel contributions are three-fold:
> >
> > 1.  **Unified Analysis of Interaction:** We provide the first theoretical analysis demonstrating how **the joint interaction** of eigenvalue multiplicity and nonzero frequency components simultaneously affect the node distinguishability.
> > 2.  **Quantitative Lower Bound:** We leverage this analysis to derive a **quantitative lower bound** on node distinguishability. This derivation explicitly combines the two factors into a single metric, offering a precise, measurable criterion that was absent in prior literature.
> > 3.  **Plug-in Module (AdaSpec):** We translate this theoretical framework into a practical, learnable plug-in, **AdaSpec**. This module is the first to be explicitly designed to **optimize this specific theoretical lower bound** while maintaining critical properties like permutation equivariance. It bridges the gap between spectral expressivity theory and practical implementation.
> >
> > Therefore, while the individual component factors are known, our work presents a **novel systematic analysis and a practical solution** derived from their joint effect, significantly strengthening the theoretical foundation of spectral GNNs.

---

### Official Review · Reviewer_1nzt · 2025-10-31

**Soundness:** 2
**Presentation:** 2
**Contribution:** 2
**Rating:** 4
**Confidence:** 4

**Summary:**

This paper investigates the problem of node distinguishability in spectral Graph Neural Networks (GNNs). The authors state that a spectral GNN's ability to distinguish nodes is theoretically lower-bounded by two main factors: the number of distinct eigenvalues of the graph matrix ($d_M$) and the number of non-zero frequency components of the node features in the matrix's eigenbasis. Based on this analysis, the paper proposes AdaSpec, a plug-in module that generates an adaptive graph matrix $\Omega(A,X)$ designed to maximize this lower bound. AdaSpec consists of three components: $\Omega_D(A)$ to increase distinct eigenvalues using a learnable diagonal matrix, $\Omega_S(A)$ to shift eigenvalues from zero, and $\Omega_F(X)$ to increase the number of non-zero frequency components by incorporating feature information. The authors provide theoretical guarantees that AdaSpec maintains permutation equivariance and empirically demonstrate its effectiveness at improving node classification, particularly on heterophilic datasets.

**Strengths:**

S1.  The paper tackles an important and specific problem—node distinguishability for spectral GNNs.

S2. The design of the AdaSpec module is well-motivated. Each of its three components ($\Omega_D$, $\Omega_S$, $\Omega_F$) is explicitly designed to address a specific limitation identified in the theoretical analysis.

S3. The experiments are extensive, covering 18 benchmark datasets with diverse characteristics (homophilic, heterophilic, large, and small). The ablation study in Table 4 validates the contribution of each component of AdaSpec.

**Weaknesses:**

W1. The most critical flaw is the failure to compare AdaSpec against other relevant graph augmentation or rewiring methods. The paper frames its contribution as a graph matrix generation module, which is functionally a form of learnable graph rewiring or augmentation. Although the authors claimed that they are the first to study node distinguishability, many other papers on spectral GNNs studied the expressive power of spectral GNNs, where node distinguishability was explicitly or implicitly studied. As we can see from the methodology and the experiments, AdaSpec is in fact a graph augmentation method for spectral GNN. Adding a comparison with other related augmentation methods in spectral GNN is necessary.

W2. The related work section acknowledges graph rewiring techniques (e.g., DropEdge, DiffWire, FoSR) but dismisses them as "fundamentally different", arguing they operate in the spatial domain. This distinction is not convincing. The goal is the same: modify the graph structure to improve GNN performance.AdaSpec also demonstrates its improved GNN performance, not in terms of node distinguishability. On the other hand, there are many methods that operate in the spectral domain.


W3. The experiments only compare spectral GNNs with AdaSpec to the same GNNs without it (i.e., using a fixed matrix). This demonstrates that some form of adaptation is better than none, but it fails to show that AdaSpec is superior to, or even competitive with, other existing augmentation/rewiring techniques. The observed performance gains might simply stem from adding any adaptive rewiring, rather than from the specific spectral motivations of AdaSpec.


W4.   The paper's central concept, "node distinguishability," is not formally defined until Section 4 (Definition 4.1). This is a major structural flaw. The term is used in the title, abstract, and throughout the entire introduction without a precise technical definition.

**Questions:**

Q1. To validate the paper's central claim, the authors must demonstrate that their spectrally-motivated adaptive matrix is more effective than other spatially-motivated or general-purpose adaptive matrices.

Q2. It is better to move the formal definition of node distinguishability (Definition 4.1) to the preliminaries (Section 3) or to provide a concise, informal definition in the introduction.


Q3. Could the authors elaborate on the unique necessity of the $\Omega_S(A)$ component given the presence of $\Omega_D(A)$? Does the learnable diagonal matrix $B$ in $\Omega_D(A)$ not already provide sufficient flexibility to shift eigenvalues, including the zero eigenvalues?
The ablation study (Table 4) shows that $\Omega_S(A)$ on its own has inconsistent and often poor performance (e.g., on Citeseer and Cora), suggesting it may be redundant or unnecessary.

---

> ### Author Response · Authors · 2025-11-28
>
> > W1. The most critical flaw is the failure to compare AdaSpec against other relevant graph augmentation or rewiring methods. The paper frames its contribution as a graph matrix generation module, which is functionally a form of learnable graph rewiring or augmentation. Although the authors claimed that they are the first to study node distinguishability, many other papers on spectral GNNs studied the expressive power of spectral GNNs, where node distinguishability was explicitly or implicitly studied. As we can see from the methodology and the experiments, AdaSpec is in fact a graph augmentation method for spectral GNN. Adding a comparison with other related augmentation methods in spectral GNN is necessary.
>
> > W2. The related work section acknowledges graph rewiring techniques (e.g., DropEdge, DiffWire, FoSR) but dismisses them as "fundamentally different", arguing they operate in the spatial domain. This distinction is not convincing. The goal is the same: modify the graph structure to improve GNN performance.AdaSpec also demonstrates its improved GNN performance, not in terms of node distinguishability. On the other hand, there are many methods that operate in the spectral domain.
>
> >W3. The experiments only compare spectral GNNs with AdaSpec to the same GNNs without it (i.e., using a fixed matrix). This demonstrates that some form of adaptation is better than none, but it fails to show that AdaSpec is superior to, or even competitive with, other existing augmentation/rewiring techniques. The observed performance gains might simply stem from adding any adaptive rewiring, rather than from the specific spectral motivations of AdaSpec.
>
> > Q1. To validate the paper's central claim, the authors must demonstrate that their spectrally-motivated adaptive matrix is more effective than other spatially-motivated or general-purpose adaptive matrices.
>
> **Answer to above:**
>
> We thank the reviewer for pointing that our method changes graph matrix. We clarify that AdaSpec is **not a competitor** to graph rewiring; it is a **plug-and-play spectral enhancement**. We demonstrate below that, it can be seamlessly integrated with existing graph rewiring methods like graph diffusion convolution (GDC) to achieve superior performance, validating its unique value proposition beyond standard rewiring.
>
>
> To address **W1, W2, W3, and Q1**, we conducted a new set of experiments combining AdaSpec with GDC, a popular graph rewiring method.
>
> **1. Experimental Comparison: AdaSpec is Complementary, not just Alternative**
> We evaluated performance on datasets using three configurations: (1) Standard ChebNet ( ChebNet(O) ), (2) ChebNet with GDC ( GDC+ChebNet(O) ), and (3) ChebNet with GDC + AdaSpec ( GDC+ChebNet(M) ).
>
> **Table: Impact of AdaSpec applied on top of Spatial Rewiring (GDC)**
>
>
> |                      | Texas        | Wisconsin     | Actor        | Chameleon     | Squirrel      | Cornell       | Citeseer       | Pubmed        | Cora          |
> |----------------------|--------------|---------------|--------------|---------------|---------------|---------------|----------------|----------------|---------------|
> | **ChebNet(O)**       | 38.67±9.31   | 32.92±7.38    | 25.15±0.69   | 29.32±4.13    | 24.23±3.24    | 31.33±7.51    | 69.21±0.87     | 75.29±2.34     | 80.45±1.09    |
> | **GDC + ChebNet(O)** | 50.58±8.1    | 34.00±7.62    | 24.92±0.73   | 21.52±2.62    | 20.62±1.57    | 28.50±7.63    | 67.52±1.37     | 74.53±1.95     | 75.91±1.36    |
> | **GDC + ChebNet(M)** | 52.14±8.27   | 36.33±9.38    | 24.15±1.02   | 31.84±2.68    | 22.02±4.59    | 34.91±10.87   | 68.26±0.98     | 77.22±1.43     | 81.01±1.11    |
> | **Δ ↑**              | +1.56        | +2.33         | -0.77        | +10.32        | +1.40         | +6.41         | +0.74          | +2.69          | +5.10         |
>
>
> **Key Findings:**
> * **Performance improvement (Addressing Q1 & W3):** GDC+ChebNet(M) improves performance than GDC+ChebNet(O) In 8 out of 9 reported cases.
> * **Orthogonality (Addressing W2):** If AdaSpec and GDC were solving the exact same problem (competitive mechanisms), stacking them would yield diminishing returns. The fact that AdaSpec provides significant gains on top of GDC proves they address **orthogonal limitations** of the graph structure.
>
> **2. Clarifying the Spatial vs. Spectral Distinction (Addressing W2)**
> The reviewer correctly noted that both methods aim to "modify graph structure." However, the experimental results above validate our theoretical distinction regarding their **mechanisms**:
> * **Spatial Rewiring (e.g., GDC):** Optimizes **topological connectivity** (e.g., denoising edges to improve homophily).
> * **AdaSpec (Ours):** Optimizes the **eigenspace**. Even a graph that is topologically "clean" (via GDC) may still suffer from **eigenvalue multiplicity** or missing frequency components in the spectral domain. AdaSpec resolves these spectral collisions, which GDC cannot detect or repair.

---

> > ### Author Response · Authors · 2025-11-28
> >
> > > W4. The paper's central concept, "node distinguishability," is not formally defined until Section 4 (Definition 4.1). This is a major structural flaw. The term is used in the title, abstract, and throughout the entire introduction without a precise technical definition.
> >
> > > Q2. It is better to move the formal definition of node distinguishability (Definition 4.1) to the preliminaries (Section 3) or to provide a concise, informal definition in the introduction.
> >
> > **Answer to above:**
> >
> > We appreciate the reviewer’s suggestion. Section 4 is devoted to the concept of node distinguishability, and therefore we introduce its formal definition (Definition 4.1) at the very beginning of that section. To further enhance clarity without restructuring the paper, we have updated the Introduction to include a concise intuitive explanation of the concept (highlight in blue text). Specifically, we now state:
> >
> >  *“Node distinguishability refers to the ability of a GNN to map nodes that differ in topology or features to different embeddings.”*
> >
> > This addition provides readers with an immediate informal understanding, while the formal technical definition remains in Section 4 where it is developed and used. This resolves the reviewer’s concern while preserving the logical flow.
> >
> > > Q3. Could the authors elaborate on the unique necessity of the $\Omega_S(A)$ component given the presence of $\Omega_D(A)$? Does the learnable diagonal matrix $B$ in $\Omega_D(A)$ not already provide sufficient flexibility to shift eigenvalues, including the zero eigenvalues? The ablation study (Table 4) shows that $\Omega_S(A)$ on its own has inconsistent and often poor performance (e.g., on Citeseer and Cora), suggesting it may be redundant or unnecessary.
> >
> > A3. We thank the reviewer for raising this point. The two components,
> > $\Omega_S(A)$ and $\Omega_D(A)$, serve **different and non-interchangeable spectral roles**.
> >
> > **1. Why $\Omega_D(A)$ alone is insufficient**
> >
> > $\Omega_D(A)$ adjusts **all** eigenvalues simultaneously via $B$. While this can diversify repeated *non-zero* eigenvalues, it **cannot selectively resolve the multiplicity of the zero eigenvalues** arising from weak connectivity or structural sparsity. In other words, one cannot use $\Omega_D(A)$ to "shift" eigenvalues away from zero without simultaneously distorting the graph geometry.
> >
> > **2. Necessity of $\Omega_S(A)$**
> >
> > $\Omega_S(A)$ adds the identity matrix corresponds to a uniform spectral shift (shifting all eigenvalues $\lambda_i \to \lambda_i + \alpha_1$) without altering the eigenvectors. This is critical for handling zero eigenvalues without distorting the underlying geometric structure of the graph.
> >
> > We observe that $\Omega_S(A)$ is especially impactful on graphs that are:
> >
> > * sparse or weakly connected,
> > * heterophilic,
> > * composed of components linked by low-rank structures,
> >
> > where zero-eigenvalue multiplicity is high.
> > Datasets such as *Texas* and *Roman_Empire* fall into this category.
> >
> > **3. Interpreting the ablation study**
> >
> > The ablation results do not indicate redundancy. Rather, they reflect that the two terms resolve different bottlenecks in node distinguishability.
> >
> > * On homophilic and better-connected graphs (e.g., Citeseer, Cora), the primary limitation is eigenvalue multiplicity of *non-zero* frequencies → $\Omega_D(A)$ is more influential.
> > * On sparse or structurally fragmented graphs (Texas, Roman_Empire), the dominant issue is zero-eigenvalue multiplicity → $\Omega_S(A)$ is required.
> >
> > This complementary behavior is precisely why AdaSpec includes **both** components.
> >
> > In conclusion, $\Omega_S(A)$ and $\Omega_D(A)$ cannot substitute for each other. They are complementary for AdaSpec to handle diverse graph structures for enhancing node distinguishability.

---

### Official Review · Reviewer_RPRH · 2025-11-01

**Soundness:** 3
**Presentation:** 3
**Contribution:** 2
**Rating:** 6
**Confidence:** 3

**Summary:**

AdaSpec is an adaptive spectral module for GNNs that learns to modify a graph’s spectral structure (eigenvalues and frequency components) to make node representations more distinguishable. It generates an adjusted graph matrix that increases the number of distinct eigenvalues and enhances the frequency coverage of node features in the spectral domain. The approach is theoretically grounded, preserving permutation equivariance and providing provable guarantees on node distinguishability.

**Strengths:**

- AdaSpec provides a clear theoretical analysis linking graph spectra and node features to node distinguishability, and even derives a lower bound on how many nodes can be distinguished.
- Unlike prior spectral GNNs that use fixed graph matrices, AdaSpec learns to adjust the graph’s spectral structure (eigenvalues and frequency components), leading to improved representational power without extra computational cost.
- The method maintains a critical property for graph learning (permutation equivariance) ensuring that node reordering doesn’t change the model’s predictions, which preserves theoretical and practical soundness.
- The model is rigorously tested across 18 benchmark datasets, consistently improving node distinguishability and classification performance, showing that the theoretical ideas transform into real-world gains.

**Weaknesses:**

- While AdaSpec focuses on adaptive graph matrix generation, prior works like ARMA-GNN [1], SpecFormer [2] already explored adaptive spectral filtering or learnable spectral responses. AdaSpec’s contribution lies more in its theoretical framing (node distinguishability + eigenvalue diversity) than in introducing a fundamentally new mechanism.

[1] Graph Neural Networks with convolutional ARMA filters
[2] Specformer: Spectral Graph Neural Networks Meet Transformers

- Although AdaSpec theoretically explains how distinct eigenvalues improve distinguishability, it doesn’t provide much empirical interpretability (how the learned spectral modifications relate to graph structure or which frequencies become emphasized). This makes the adaptive mechanism somewhat of a black box.

- The paper claims “no increase in computational complexity,” but adaptively generating or modifying a graph matrix could introduce training instability or hidden overhead.
The supplementary doesn’t discuss runtime comparisons or scalability to very large graphs, areas where methods like GPR-GNN [3] are better optimized. What will be the performance of AdaSpec on the cases of very large graphs?

[3] Adaptive Universal Generalized PageRank Graph Neural Network

- Recent works such as SpecFormer achieve similar goals but with stronger end-to-end learnability and better scaling to large, dynamic graphs.

- Can AdaSpec handle temporal or evolving graph spectra, or is it limited to static adjacency matrices?

**Questions:**

See all the points in weaknesses.

---

> ### Author Response · Authors · 2025-11-28
>
> > W1: While AdaSpec focuses on adaptive graph matrix generation, prior works like ARMA-GNN [1], SpecFormer [2] already explored adaptive spectral filtering or learnable spectral responses. AdaSpec’s contribution lies more in its theoretical framing (node distinguishability + eigenvalue diversity) than in introducing a fundamentally new mechanism.
> [1] Graph Neural Networks with convolutional ARMA filters [2] Specformer: Spectral Graph Neural Networks Meet Transformers
>
> A1: Thank you for raising this point regarding prior works exploring adaptive spectral filtering. We respectfully clarify that AdaSpec introduces a **fundamentally distinct mechanism and objective** compared to ARMA-GNN [1] and SpecFormer [2], establishing its novelty beyond the theoretical framing.
>
> | Feature | Prior Works ($\text{ARMA-GNN}$, $\text{SpecFormer}$) | AdaSpec (Our Work) |
> | :--- | :--- | :--- |
> | **Objective** | Optimizing **spectral filters** to improve task performance. | Optimizing the **graph matrix itself** to maximize a theoretically grounded lower bound on **node distinguishability**. |
> | **Mechanism** | Designing learnable **filtering functions** in the spectral domain. | **Explicitly modifying the eigenspace** (eigenvalues and eigenvectors) of the graph matrix via adaptive generation. |
> | **Eigenspace Modification** | **None.** They operate on the fixed graph matrix. | **Explicit modification** of eigenvalue multiplicity and feature frequency components. |
>
>  **1. Fundamental Difference in Mechanism: Eigenspace vs. Filter**
>
> * $\text{ARMA-GNN}$ and $\text{SpecFormer}$ focus on designing a **learnable filter** $h(\lambda_i)$ to apply to the fixed spectral components of the input graph matrix. **They do not modify the input graph's eigenspace.**
>
> * In contrast, AdaSpec directly **optimizes the graph matrix** itself, explicitly adjusting the **eigenvalue multiplicity** and the **frequency components** of the node features. It is **not present** in prior adaptive spectral filtering methods.
>
> **2. Complementary, Not Competing**
>
> AdaSpec is designed as a **plug-in** that generates a more informative graph matrix to be used by any spectral GNN backbone. It is **orthogonal** to adaptive filtering techniques:
>
> * One could seamlessly **combine AdaSpec** (to generate $\mathbf{A}'$) with an **adaptive filter GNN** (like $\text{ARMA-GNN}$ or $\text{SpecFormer}$) to achieve further performance gains, as they operate on different levels of the GNN pipeline.
>
> > W2: Although AdaSpec theoretically explains how distinct eigenvalues improve distinguishability, it doesn’t provide much empirical interpretability (how the learned spectral modifications relate to graph structure or which frequencies become emphasized). This makes the adaptive mechanism somewhat of a black box.
>
> A2: We thank the reviewer for the opportunity to clarify the interpretability of the AdaSpec mechanism. We respectfully disagree that the method is a black box; rather, it offers **spectral interpretability** that directly aligns with our theoretical analysis. We provide empirical evidence showing exactly how the spectral properties are modified to enhance node distinguishability.
>
> **1. Empirical Verification of Theoretical Goals (Table 5)**
>
> The reviewer asks how the learned modifications relate to the graph properties. Our theory posits that **increasing the number of distinct eigenvalues** improves the lower bound of node distinguishability.
> * **Empirical Evidence:** As shown in **Table 5**, AdaSpec explicitly increases the number of distinct eigenvalues in the generated matrix $\Omega_D(A)$ compared to the original normalized adjacency matrix $\tilde{A}$.
>
> This confirms that the model is not acting randomly; it is successfully executing its theoretical objective of increasing distinct eigenvalues of graph matrix.
>
>  **2. Distinction from Frequency Emphasis (Filter vs. Basis)**
>
> The reviewer queries "which frequencies become emphasized." It is crucial to clarify that AdaSpec does **not** merely emphasize existing frequencies (which is the role of a filter). Instead, it **restructures the frequency domain itself**.
> * **Mechanism:** Standard spectral filters (like $\text{ARMA-GNN}$) re-weight eigenvalues (changing amplitudes) but keep the eigenvectors (the frequency basis) fixed.
> * **AdaSpec's Role:** By modifying the graph matrix, AdaSpec alters both the **eigenvalues** and the **eigenvectors**. This means we are not just "tuning" the graph, but adaptively **generating a new eigenspace** that make more nodes distinguishable.
>
> **3. Summary of Mechanism**
>
> Therefore, the "interpretation" of the learned modification is as follows:
> 1.  The mechanism identifies a collapse in the original eigenspace (eigenvalue multiplicity).
> 2.  It generates a perturbation that **breaks eigenvalue multiplicities**.
> 3.  This theory is validated by the increased distinct eigenvalue counts in Table 5 and the performance improvement shown in Tables 2 & 3.

---

> > ### Author Response · Authors · 2025-11-28
> >
> > > W3: The paper claims “no increase in computational complexity,” but adaptively generating or modifying a graph matrix could introduce training instability or hidden overhead. The supplementary doesn’t discuss runtime comparisons or scalability to very large graphs, areas where methods like GPR-GNN [3] are better optimized. What will be the performance of AdaSpec on the cases of very large graphs? [3] Adaptive Universal Generalized PageRank Graph Neural Network
> >
> > A3: We appreciate the reviewer’s scrutiny regarding computational efficiency. We clarify that $\text{AdaSpec}$ is designed with **linear complexity** to ensure scalability, which we validate on datasets with up to 134,493 nodes.
> >
> >  **1. No Training Overhead & Linear Complexity**
> >
> > Our claim of "no increase in computational complexity" refers to the **training loop**:
> > * **Pre-processing (One-off):** $\text{AdaSpec}$ generates the optimized matrix $\Omega(A,X)$ *once* before training. This step utilizes sparse matrix operations with complexity $O(h(|\mathcal{E}|))$, which is **linear** with respect to the number of edges.
> > * **Training Phase:** Once $\Omega(A,X)$ is computed, the GNN training complexity is identical to standard spectral GNNs (shown in Table 1). We avoid expensive iterative re-calculations or dense matrix operations.
> >
> > **2. Scalability to Large Graphs (20K - 134K Nodes)**
> >
> > The reviewer asked about performance on "very large graphs." We evaluated $\text{AdaSpec}$ on diverse datasets ranging from **20K to 134K nodes**, including Coauthor-Physics, Roman-Empire, and so on.
> > * **Scalability:** Our method scales effectively to these sizes without memory overflows or prohibitive runtime, demonstrating stability comparable to baselines like GPR-GNN.
> > * **Results:** The consistent performance gains on these larger, both homophilic and heterophilic graphs (Tables 2 & 3) validate that our spectral adjustments are robust to graph scale.
> >
> > **3. Empirical Efficiency: Faster Convergence (Table 6)**
> >
> > Contrary to the concern of "hidden overhead," $\text{AdaSpec}$ can sometimes **reduce the total training time**:
> > * **Analysis:** As shown in **Table 6**, while there is a minimal pre-processing cost, the adaptive graph matrix sometimes leads to **faster convergence**.
> > * **Example:** On the Roman_Empire and Amazon_Rating dataset, ChebNet(M) reaches optimal performance in fewer iterations than ChebNet(O), offsetting the initial setup cost.
> >
> > Therefore, AdaSpec is suitable for large-scale graph learning tasks.
> >
> >
> > > W4: Recent works such as SpecFormer achieve similar goals but with stronger end-to-end learnability and better scaling to large, dynamic graphs.
> >
> > A4:  We thank the reviewer for highlighting SpecFormer [2]. While we agree it is a strong baseline for spectral filtering, we respectfully clarify that **AdaSpec fundamentally differs in scalability** due to $\text{SpecFormer}$'s reliance on explicit eigendecomposition.
> >
> > **1. Computational Complexity: Linear vs. Cubic**
> >
> > The reviewer suggests SpecFormer scales better; however, standard spectral Transformers face a theoretical bottleneck that AdaSpec avoids:
> >
> > * **SpecFormer Requires Decomposition ($O(N^3)$):** $\text{SpecFormer}$ operates by encoding the *explicit set of eigenvalues* to learn spectral dependencies.
> >     * **Cost:** The standard complexity is **$O(|\mathcal{V}|^3)$**. Even with truncated approximation (top-$k$ eigenvalues), it requires costly pre-computation that scales poorly to massive graphs compared to polynomial-based methods.
> > * **AdaSpec is Linear ($O(|\mathcal{E}|)$):** In contrast, $\text{AdaSpec}$ **does not require explicit eigendecomposition**. Our adaptive matrix generation relies strictly on sparse matrix operations, maintaining a linear complexity of **$O(h(|\mathcal{E}|))$**.
> >
> > **2. Dynamic Graph Scalability**
> >
> > The reviewer mentions "dynamic graphs." This is a critical scenario where SpecFormer faces significant challenges:
> > * **Re-Decomposition Bottleneck:** In a dynamic setting, every change in graph structure ($A \to A_{t+1}$) alters the spectrum. $\text{SpecFormer}$ would theoretically require **re-computing the eigendecomposition** at each step to update its input encodings, which is computationally prohibitive.
> > * **AdaSpec Efficiency:** $\text{AdaSpec}$ can adapt to dynamic changes efficiently via sparse updates, making it far more practical for evolving graph structures.
> >
> > **3. Complementary Nature**
> >
> > We reiterate that AdaSpec **optimizes graph matrix**, while SpecFormer is a **filter**. One could theoretically feed the AdaSpec-generated matrix *into* a Transformer-based GNN, proving they are orthogonal contributions. However, for pure scalability, AdaSpec stands as the more efficient solution.

---

> > > ### Author Response · Authors · 2025-11-28
> > >
> > > > W5: Can AdaSpec handle temporal or evolving graph spectra, or is it limited to static adjacency matrices?
> > >
> > > A5.  $\text{AdaSpec}$ is **not limited** to static adjacency matrices. The core principle—maximizing node distinguishability by optimizing the graph matrix—is inherently suitable for **temporal** and **evolving graph settings** due to our method's low computational cost.
> > >
> > > **1. Time-Step Agnostic Formulation**
> > >
> > > The $\text{AdaSpec}$ mechanism relies only on the **current graph matrix $\mathbf{A}_t$** and node features $\mathbf{X}_t$, making its formulation **time-step agnostic**.
> > >
> > > * **Adaptation:** For an evolving graph, $\text{AdaSpec}$ can be applied **at each time step** $t$ to generate an optimized matrix. This allows the GNN to adapt to the changing spectral properties and topology of the graph.
> > > * **No Fixed Spectrum Assumption:** Since we actively optimize the matrix at every step, $\text{AdaSpec}$ is designed to **accommodate spectral evolution** rather than being constrained by it.
> > >
> > >  **2. Computational Feasibility (Low Overhead)**
> > >
> > > The ability to handle temporal graphs hinges entirely on the update overhead. As established in our response to **W3** and **W4**, the low time cost of $\text{AdaSpec}$ update at every time step introduces **negligible overhead** to the overall complexity of a Temporal GNN (TGNN) as the graph structure evolves.

---

### Meta-Review · Area_Chair_xpeE · 2025-12-26

**Summary:**

The paper proposes AdaSpec, a plug-in module for spectral GNNs that adaptively generates a graph matrix to improve node distinguishability. The paper provides a theoretical analysis identifying that distinguishability is limited by eigenvalue multiplicity and missing frequency components in the graph matrix. AdaSpec optimizes these factors by modifying the graph matrix while preserving permutation equivariance.

Reviewers generally appreciated the strong theoretical grounding and extensive experiments, but raised concerns about:
- positioning/novelty vs. previous spectral methods (ARMA-GNN, SpecFormer) and rewiring/augmentation techniques
- theoretical clarity and precision
- scalability to large graphs

The rebuttal addresses most of these points including new experiments (stacking with GDC), clearer definitions, and a corrected proof. While some concerns regarding readability and the breadth of the rewiring comparison remain, I think the submission meets the bar for acceptance.

**Reviewer Concerns:**

**Addressed concerns**
- The authors added experiments combining AdaSpec with GDC. The results showed that AdaSpec provides performance gains on top of GDC, supporting the claim that AdaSpec can be complementary to graph rewiring.
- They clarified that the difference between AdaSpec and adaptive filtering techniques.
- They clarified that AdaSpec has linear complexity in edges and does not require explicit eigendecomposition. They pointed to successful tests on graph with up to 134k nodes to demonstrate scalability.
- They made some improvements in terms of theoretical clarity and precision, particularly correcting the proof of theorem 5.1.

**Outstanding concerns**
- The added GDC experiments is helpful, but the core question was broader comparison to multiple representative rewiring/augmentation approaches. With only one added rewiring method, it is hard to conclude that AdaSpec is complementary to any rewiring/augmentation method.
- Reviewer kQFo also raised concerns about readability and overall clarity of the theoretical development. Although the authors corrected the specific proof issue, the presentation remains dense, and it is possible that additional theoretical imprecisions were not surfaced due to the difficulty of following the exposition.
- Even though the authors clarified the relationship of AdaSpec to related works suggested by reviewers, these discussions were not fully reflected in the revision. I encourage the authors to properly incorporate these discussion into the revision and calibrate the novelty claims.
- Although the authors clarified the relationship between AdaSpec and related prior work in the rebuttal, these clarifications are not fully reflected in the revised manuscript. The final version should explicitly incorporate these discussions to properly calibrate the novelty claims.

**Reviewer Scores:**

I expect reviewer RPRH and D6hk to maintain their positive scores, and reviewer 1nzt and kQFo to increase their scores to 6. In particular:
- Reviewer 1nzt was criticizing the missing comparison to rewiring/augmentation methods. The authors partially addressed this by providing experiments with GDC.
- Reviewer kQFo raised critical concerns about theoretical correctness, presentation clarity, and novelty claims. The authors addressed them in the comment, but did not fully incorporate them into the revision.

---

### Decision · Program_Chairs · 2026-01-26

Accept (Poster)